# Comprehensive characterisation of IAA inactivation pathways reveals the impact of glycosylation on auxin metabolism and plant development in Arabidopsis

Rubén Casanova-Sáez[1], Aleš Pěnčík[1,2,3], Federica Brunoni [2,3,4], Anita Ament [2,3], Pavel Hladík[2,3], Asta Žukauskaitė[5], Jan Šimura[1], Ute Voß[6], Ondřej Novák [2,3], Malcolm Bennett[6], Karin Ljung [1] ✉ & Eduardo Mateo-Bonmatí [1,7] ✉

Alongside biosynthesis and transport, inactivation regulates indole-3-acetic acid (IAA) concentration, a compound with numerous functions in plant development. The main inactive IAA metabolites are oxidised forms and ester- or amide-linked conjugates. DIOXYGENASE FOR AUXIN OXIDATION1 (DAO1) and DAO2, 2-oxoglutarate and iron-dependent dioxygenases, contribute to IAA oxidative inactivation with group II GRETCHEN HAGEN3 (GH3) IAA-amido synthetases, while UDP-glycosyltransferases (UGTs) conjugate IAA to sugars. To study IAA inactivation routes, we generated combinatorial Arabidopsis mutants between all group II *GH3s* (*gh3oct*) and *DAO1* or *DAO2*, and between *DAOs* and the main *UGTs*. In vivo [$^{13}C_6$]IAA feeding experiments traced exogenously applied IAA's metabolic fate, supporting the main IAA inactivation pathway where DAOs act downstream of GH3s. Results from these experiments also indicated UGT-mediated IAA glycosylation is more important than previously assumed for modulating IAA levels and plant development. Our metabolic and transcriptomic data revealed that *gh3oct* may still retain some GH3 activity, explaining previously reported phenotypic inconsistencies. Our data additionally suggest that unidentified metabolic activities might play a role in IAA overproducing plants, and that premature downregulation of flowering time regulators like *FLOWERING LOCUS C* (*FLC*) likely underlies early flowering of *gh3oct* and *gh3oct dao1-6* plants.

As biochemical messengers, hormones synchronize developmental stages with environmental conditions to ensure the survival and reproduction of eukaryotic organisms. This is particularly important in the plant lineage since their sessile nature hampers other possible responses. Among plant hormones, indole-3-acetic acid (IAA), the most important form of auxin, has been found to play a role in many developmental transitions[1,2] as well as in the integration of external signals[3,4] and stress responses[5,6]. Auxin´s action depends on its cellular concentration. Contrary to the transport-centric original dogma[7,8], it is now believed that local auxin concentration results from concerted action of at least three convergent and intertwined pathways: auxin (polar) transport, auxin biosynthesis, and auxin inactivation[1,9]. All of them are complex pathways with multiple genes and cross-regulatory relationships.

IAA inactivation involves modifications of its molecular structure, rendering it unrecognizable to the signalling and transport machinery. Several chemical modifications have been observed on the IAA molecule.

[1]Umeå Plant Science Centre, Department of Forest Genetics and Plant Physiology, Swedish University of Agricultural Sciences, Umeå, Sweden. [2]Laboratory of Growth Regulators, Faculty of Science, Palacký University, Olomouc, Czech Republic. [3]Laboratory of Growth Regulators, Institute of Experimental Botany, The Czech Academy of Sciences, Olomouc, Czech Republic. [4]Department of Biotechnology, University of Verona, Strada le Grazie 15, 37134 Verona, Italy. [5]Department of Chemical Biology, Faculty of Science, Palacký University, Olomouc, Czech Republic. [6]Division of Plant and Crop Sciences, School of Biosciences, University of Nottingham, Sutton Bonington Campus, Loughborough, UK. [7]Centro de Biotecnología y Genómica de Plantas (CBGP), Universidad Politécnica de Madrid (UPM), Instituto Nacional de Investigación y Tecnología Agraria y Alimentaria (INIA)/CSIC, Pozuelo de Alarcón, Madrid, Spain. ✉e-mail: Karin.ljung@slu.se; eduardo.mateo@upm.es

IAA can be inactivated by methylation (MeIAA)[10] and via ester-linked and amide-linked conjugation[1]. The most abundant inactive form is the ester-linked IAA-glucose (IAA-glc), present at high levels in different plants, especially in seedlings and seeds[11–13]. Enzymes catalysing the glycosylation of IAA belong to the family of UDP-glycosyltransferases. Despite being a large family, only three UGTs have been identified as playing a role in IAA inactivation in Arabidopsis: UGT84B1, UGT74D1, and UGT76E5[14–16]. Amide-linked conjugates comprise a group of molecules where IAA is conjugated mainly to an amino acid (IAA-aa) or to small peptides and proteins. The formation of IAA-aa conjugates is catalysed by the GRETCHEN HAGEN3 (GH3) family of acyl acid amido synthetases[17], another big gene family containing 19 members in Arabidopsis, clustered in three functional groups[18]. Group II GH3 members are known to catalyse the formation of the different IAA-aa conjugates, which include IAA-leucine, IAA-alanine, IAA-phenylalanine, IAA-aspartate (IAA-Asp), or IAA-glutamate (IAA-Glu)[19]. Oxidized counterparts of these IAA-aa conjugates, such as 2-oxindole-3-acetic acid-aspartate (oxIAA-Asp) and oxIAA-Glu are also formed[20,21].

Another inactive IAA is found in the oxidized form (oxIAA), initially reported to be generated by the activity of the 2-oxoglutarate and iron-dependent dioxygenases (2OGD) DIOXYGENASE FOR AUXIN OXI-DATION 1 (DAO1) and DAO2[22–24]. Recent reports, however, indicate that DAOs may operate mainly downstream of GH3s by converting IAA-aa conjugates into oxIAA-aa conjugates[20,25], later transformed into oxIAA by the action of amidohydrolases like IAA-LEUCINE RESISTANT1 (ILR1)[25].

To elucidate the role of the main auxin inactivation routes (Fig. 1) in regulating IAA levels and plant development, we generated a comprehensive set of genetic mutants. We examined phenotypic outcomes and quantified the metabolic fate of [$^{13}C_6$]-labelled IAA across all genotypes to trace the activity of distinct inactivation routes and their associated developmental implications. We also conducted transcriptomic analyses of the IAA response in two genotypes impaired in the primary inactivation mechanisms. Our results outline an important role of the IAA glycosylation pathway in IAA metabolism and plant development, and support the notion that GH3s and DAOs act in a primary and consecutive pathway for IAA inactivation. Finally, our findings suggest the existence of additional, yet unidentified, players contributing to IAA inactivation.

## Results
### Generation of a genetic toolkit to study IAA inactivation
To investigate the contribution of the major IAA inactivation pathways (Fig. 1) to plant development, we undertook a reverse genetic approach to generate mutants that disrupt these pathways individually and in combination. For this purpose, we combined previously generated lines with newly generated CRISPR/Cas9-induced mutants. Previously, we identified UGT84B1 and UGT74D1 as the main UDP-glycosyltransferases responsible for conjugating glucose to both IAA and oxIAA[14]. To effectively disrupt the IAA glycosylation pathway, we generated a ugt84b1 ugt74d1 double mutant by crossing the CRISPR/Cas9-mediated knock-out of UGT84B1[14] in the T-DNA ugt74d1 mutant background[15].

In parallel, we generated a dao1 dao2 double mutant by introducing a CRISPR/Cas9-mediated deletion of DAO1 in the dao2-1 insertion allele background[23]. This deletion (hereafter referred to as dao1-4) removed a 712 bp genomic fragment of within DAO1, resulting in a frameshift change at the amino acid 75 and a premature stop codon at the amino acid 87. This truncation effectively eliminates the 2-oxoglutarate and iron-dependent dioxygenase (2OGD) domain, essential for enzymatic function, from the resulting mutant protein (Fig. 2).

To simultaneously disrupt the IAA glycosylating and oxidative pathways, we used CRISPR/Cas9 to induce a 2.5-kb genomic deletion encompassing both DAO1 and DAO2 genes in the ugt74d1 background. This deletion removed the 2OGD domains from both DAO1 and DAO2 (Fig. 2; dao1-5 dao2-6). The resulting deletion was then combined with the ugt84b1 ugt74d1 double mutant by crossing, yielding the triple mutant ugt84b1 dao1-5 dao2-6, as well as the quadruple mutant dao1-5 dao2-6 ugt84b1 ugt74d1.

**Fig. 1 | Main pathways of indole-3-acetic acid (IAA) metabolic inactivation in Arabidopsis.** Along with biosynthesis, transport, and subcellular compartmentalization, inactivation defines cellular IAA levels. IAA metabolites are indicated in circular boxes. Enzymes catalyzing metabolic reactions are indicated in magenta. IAA-Glc: IAA-glucose (indole-3-acetyl-β-D-glucopyranoside); IAA-aa: IAA–amino acid conjugates, such as IAA-Asp (indole-3-acetyl-L-aspartic acid) and IAA-Glu (indole-3-acetyl-L-glutamic acid); oxIAA-aa: conjugates of oxIAA (2-oxoindole-3-acetic acid) with amino acids; oxIAA-Glc: oxIAA-glucose (2-oxoindole-3-acetyl-β-D-glucopyranoside). GH3s: GRETCHEN HAGEN3 IAA acyl acid amido synthetases. DAO1/2: DIOXYGENASE FOR AUXIN OXIDATION 1/2. ILR1/ILLs: IAA-LEUCINE RESISTANT1/ILR1-LIKE enzymes. UGT84B1/74D1/76E5: UDP-glucosyltransferases 84B1, 74D1, and 76E5.

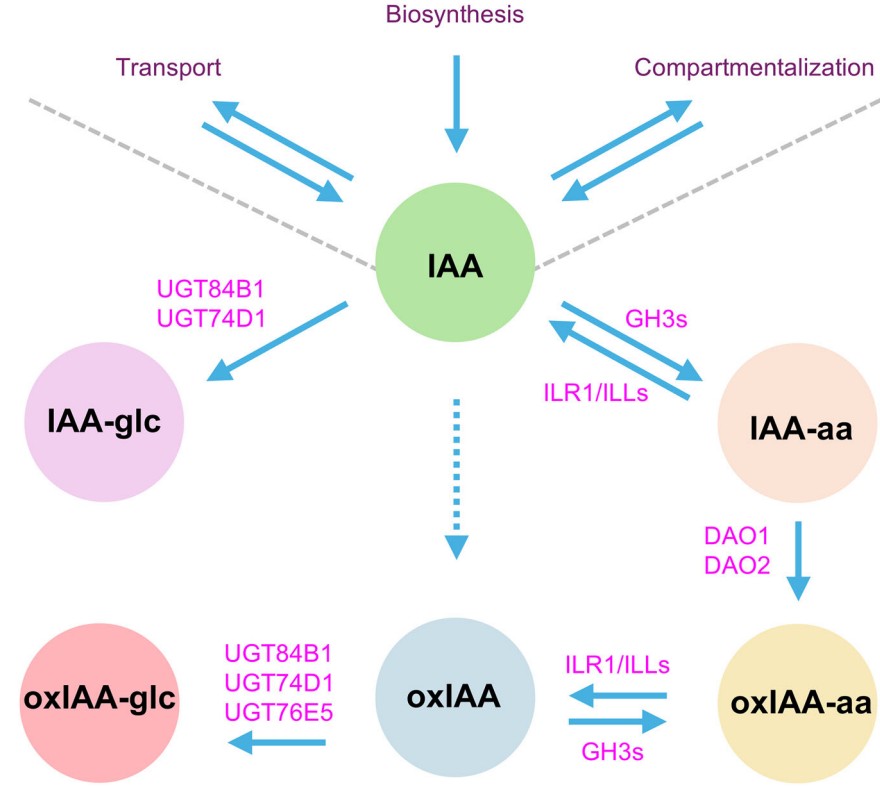

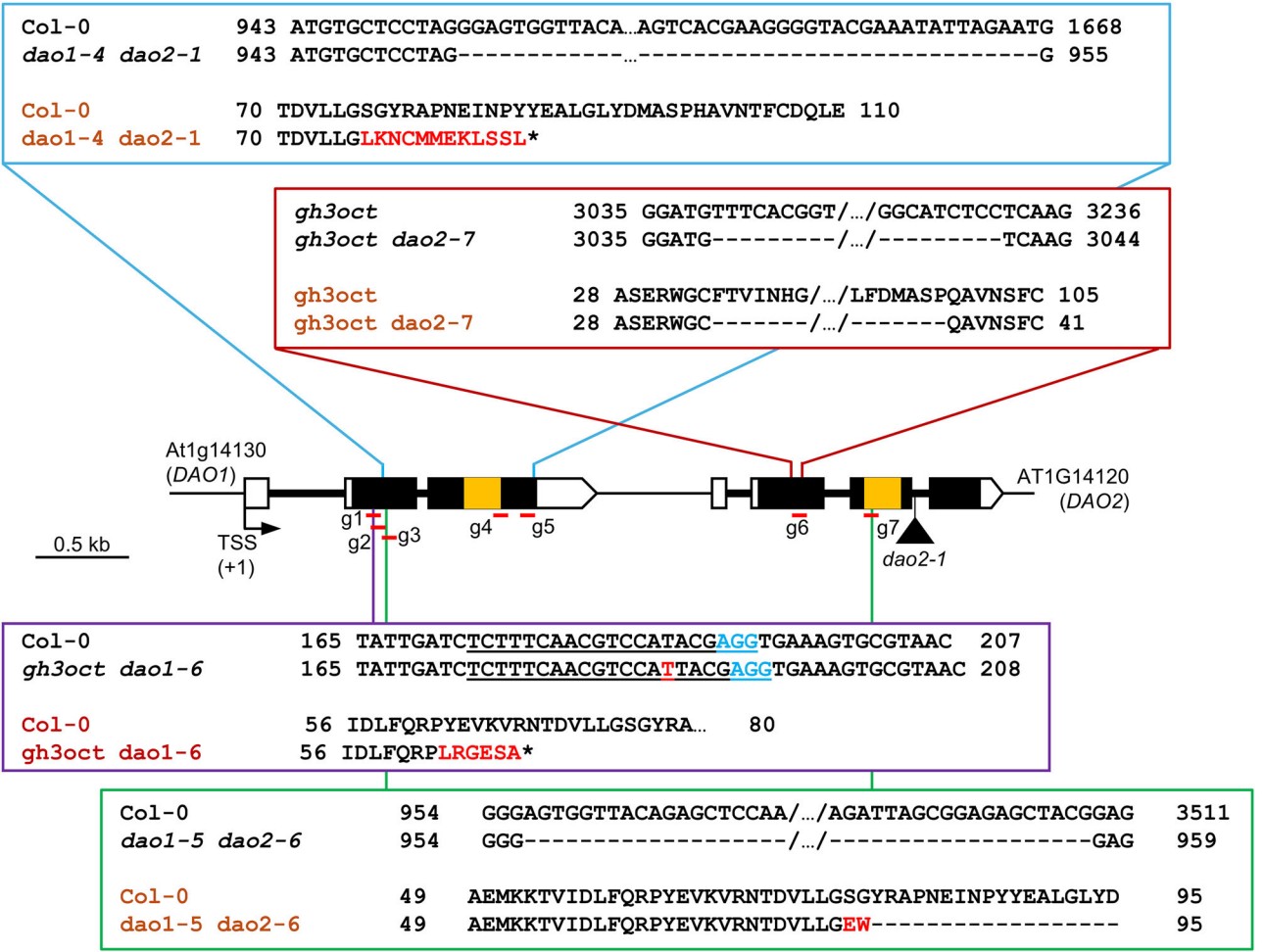

**Fig. 2 | Schematic representation of the multiple mutants generated.** Illustration of editing events obtained after targeting the CRISPR/Cas9 system to the genomic region encompassing the *DAO1* and *DAO2* genes. Gene architecture of *DAO1* and *DAO2* and illustration of the nature and location of the mutations analyzed in this work. Black boxes indicate exons and lines indicate introns. White boxes represent untranslated regions. The triangle represents a T-DNA insertion (*dao2-1*). Orange stripes represent the region encoding the 2-oxoglutarate/Fe(II)-dependent dioxy-genase domain (2OGD; IPR005123). Red horizontal bars represent sgRNAs (g1–g7; not drawn to scale) used to edit the region. The blue box magnifies the DAO1

deletion found in the *dao2-1* genetic background. Genotypes in black refer to genomic regions, whereas genotypes in brown refer to proteins. The purple box illustrates the single-nucleotide variant causing a premature stop codon in the *gh3oct* background. The NGG protospacer adjacent motif (PAM) is highlighted in light blue. The green box shows the large genomic deletion encompassing *DAO1* and *DAO2* in the *ugt74d1* background. Red letters represent nucleotides or amino acids absent in the wild type. Asterisks indicate premature stop codons. Numbers adjacent to sequences indicate nucleotide position relative to the transcriptional start site (TSS) of *DAO1* or amino acid position. Scale bar indicates 0.5 kb.

To examine the interaction between the conjugation and oxidation pathways for IAA inactivation, we further disrupted either *DAO1* or *DAO2* in the *gh3.1 gh3.2 gh3.3 gh3.4 gh3.5 gh3.6 gh3.9 gh3.17* octuple mutant (*gh3oct*)[19], generating the *gh3oct dao1-6* and *gh3oct dao2-7* nonuple mutants. The isolated mutation in *DAO1* consisted of a single-nucleotide insertion at position 187 of the *DAO1* transcriptional unit, resulting in a frameshift and a premature stop codon after residue 68 (Fig. 2). The *dao2-7* allele was generated via CRISPR/Cas9-mediated deletion of 192 nucleotides, resulting in the loss of 64 amino acids from the DAO2 protein (Fig. 2). Together, these lines constitute a comprehensive set of combinatorial mutants to dissect the roles of the IAA inactivation pathways in auxin metabolism and plant development.

**Phenotypic analyses reveal unique contributions of IAA glyco-sylases and DAO2 to plant development**

We then carried out phenotypic analyses of the complete set of generated mutant lines. Consistent with previous reports, *dao1-1* and *dao2-1* single mutants exhibited virtually no root phenotype or hypocotyl phenotype (Fig. 3a-c)[22,23]. In contrast, *dao1-4 dao2-1* double mutants displayed a mild,

high auxin-associated phenotype of increased primary root length (PRL; Figs. 3a; S1). This reveals DAO1 and DAO2 having at least partially over-lapping functions and being able to compensate for each other's loss, despite DAO1 being the predominant oxidase in root IAA metabolism[22].

While single *ugt* mutants have previously been reported to lack apparent root phenotypic alterations[14] (Fig. S1b-g), *ugt84b1 ugt74d1* double mutant seedlings exhibited increased hypocotyl length (Fig. 3c). This indicates that IAA glycosylation, redundantly mediated by UGT84B1 and UGT74D1, plays a significant role in modulating auxin levels in the hypocotyl during seedling development. Remarkably, the lateral root den-sity (LRD) in the quadruple mutant *ugt74d1 ugt84b1 dao1-5 dao2-6* mutant was synergistically increased when compared to *dao1 dao2*, and *ugt74d1 ugt84b1* double mutants (Fig. 3b), which suggests that IAA oxidation and glycosylation pathways redundantly contribute to auxin homeostasis and root architecture, while being individually dispensable under standard growth conditions.

In line with the literature, knocking out the *GH3* pathway had more severe developmental impacts compared to *dao* or *ugt* double mutants. The *gh3oct* mutant showed enhanced hypocotyl length and LRD without penalty

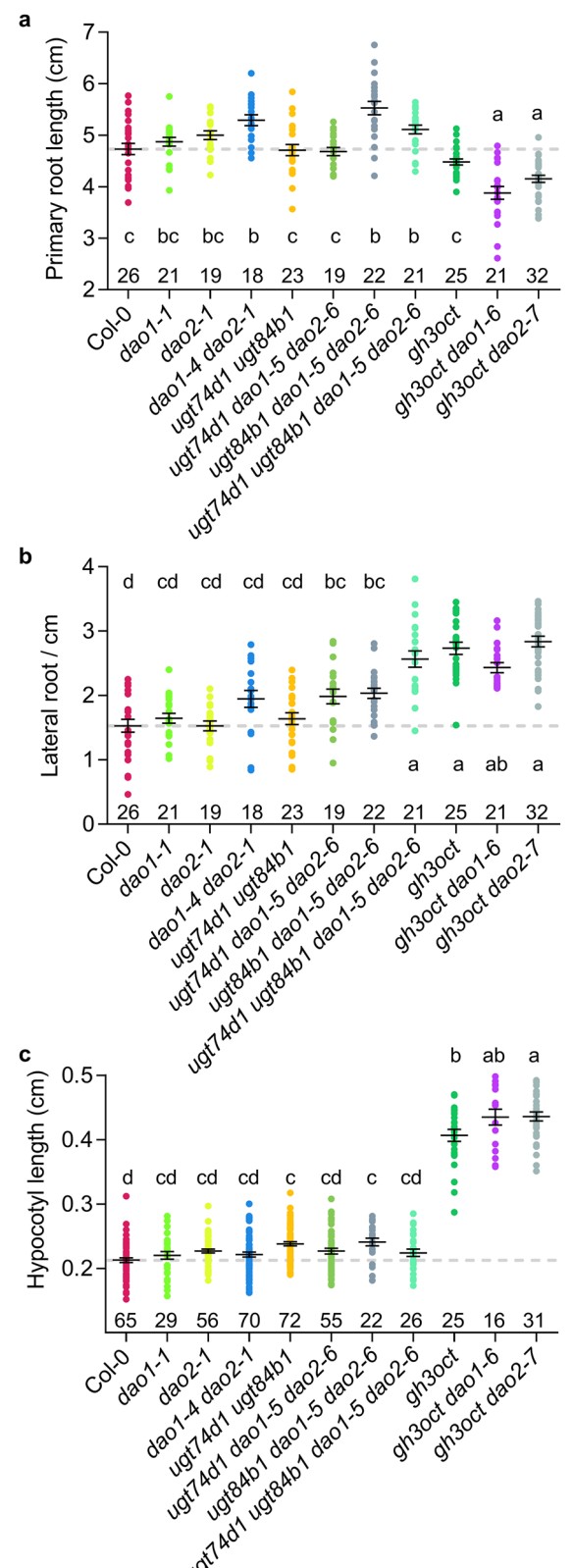

**Fig. 3 | Morphometric traits of IAA metabolism mutants. a–c** Phenotypic variables measured across the different genotypes. Dots represent individual measurements of (**a**) primary root length, **b** lateral root density, and **c** hypocotyl length. Traits in (**a**) and (**c**) were measured 7 days after stratification (DAS), whereas (**b**) was measured at 10 DAS. Mean ± standard error of the mean is shown. Grey dashed lines indicate the wild-type mean. A one-way ANOVA was performed followed by pairwise comparisons using Tukey's HSD test. Conditions marked with different letters (**a–c**) are significantly different ($p < 0.05$). The number of plants analyzed per genotype is shown above each genotype name.

## IAA-glycosylation deficient mutants show significant re-wiring of IAA inactivation routes

To precisely dissect the contributions of the different enzyme sets to IAA metabolism, we quantified the levels of $^{13}C_6$-isotope-labelled IAA metabolites, including $[^{13}C_6]$IAA, $[^{13}C_6]$IAA-glc, $[^{13}C_6]$IAA-Glu, $[^{13}C_6]$IAA-Asp, $[^{13}C_6]$oxIAA, $[^{13}C_6]$oxIAA-glc, $[^{13}C_6]$oxIAA-Glu, and $[^{13}C_6]$oxIAA-Asp in 7-day-old seedlings from all generated mutant lines after feeding with $[^{13}C_6]$IAA for 3, 12, and 24 h (Figs. 4 and S2). In parallel, we also measured the steady-state levels of these metabolites in root tissues across genotypes (Fig. S3).

Single *dao1-1* mutant reproduced previously reported metabolic profiles, with a massive accumulation of IAA-aa conjugates[22,23] and impaired formation of the oxidised forms $[^{13}C_6]$oxIAA, $[^{13}C_6]$oxIAA-glc, $[^{13}C_6]$oxIAA-Glu or $[^{13}C_6]$oxIAA-Asp (Fig. S2). In contrast, the IAA metabolic profile of *dao2-1* closely resembled that of the wild type (Fig. S2). Both de novo formation of $[^{13}C_6]$IAA-derived inactive metabolites and the steady-state levels of endogenous IAA metabolites in seedling roots from the *dao1-4 dao2-1* double mutant were comparable to those observed in *dao1-1* (Fig. 4, S2, and S3). Interestingly, despite being reduced relative to Col-0, steady-state oxIAA, oxIAA-glc, and oxIAA-aa were still present in both single *dao1-1* and double *dao1-4 dao2-1* mutant plants (Fig. S3e-g). One explanation for this phenomenon could be that *DAO2* compensates the lack of *DAO1* in both the *dao1-1* single and the *dao1-1 dao2-1* double, since it was reported that *dao2-1* is a leaky allele of *DAO2*[23]. However, our phenotypic analyses rather disfavour this possibility; other, more deleterious *DAO2* alleles of were obtained in combination with *ugt* mutants without any enhancement of the effects on root growth (Fig. 3). Moreover, these additional *dao2* alleles generated in this work show the same feeding (Fig. 4) and steady-state levels (Fig. S3) of oxIAA, IAA-Glu, IAA-Asp, or oxIAA-glc, suggesting the involvement of non-catalytic oxidation processes and/or the existence of additional, yet unidentified, oxidase activities contributing to IAA turnover.

Not unexpectedly, the formation of $[^{13}C_6]$IAA-glc was strongly impaired in the *ugt74d1 ugt84b1* mutant (Fig. 4b). The statistically significant increase in the formation of other inactive metabolites $[^{13}C_6]$oxIAA, $[^{13}C_6]$IAA-Glu, and $[^{13}C_6]$IAA-Asp (Fig. 4c-e) suggests a re-wiring of accumulating IAA towards the oxidative and GH3 pathway as a consequence of disrupting IAA glycosylation. While blocking both the IAA oxidative and glycosylation pathways in the quadruple *ugt84b1 ugt74d1 dao1 dao2* mutant had a stronger impact on lateral root density compared to doubles and triples (Fig. 3b), the IAA inactivation network and endogenous levels of inactive IAA forms were similar between the quadruple *ugt84b1 ugt74d1 dao1 dao2* and the double *ugt84b1 ugt74d1* or *dao1 dao2* mutants (Fig. 4 and S3). Notably, our data showed identical reduction in $[^{13}C_6]$IAA accumulation and endogenous IAA levels in both the double *ugt74d1 ugt84b1* and the quadruple *ugt84b1 ugt74d1 dao1 dao2* (Figs. 4A and S2A), seemingly resulting from the loss of UGT74D1 (Figs. S2A and S3A). This indicates that an impaired IAA glycosylation pathway triggers enhanced metabolic removal of IAA, likely via the GH3 pathway (Fig. 4c, d, g, h) or by additional IAA inactivation routes yet to be discovered.

The triple mutant *ugt74d1 dao1 dao2* showed another interesting IAA metabolic rewiring. While, as expected, it showed reduced steady-state levels of oxIAA, lack of oxIAA-glc, and a substantial accumulation of IAA-Glu

in the primary root length (Fig. 3a-c). Notably, the *gh3oct dao1-6* and *gh3oct dao2-7* nonuple mutants displayed a further reduction in PRL and a greater increase in hypocotyl length compared to *gh3oct* alone (Fig. 3b,c), which is indicative of higher IAA levels in nonuple mutant plants and again supports an overlapping yet non-fully redundant role of DAO2 in plant development.

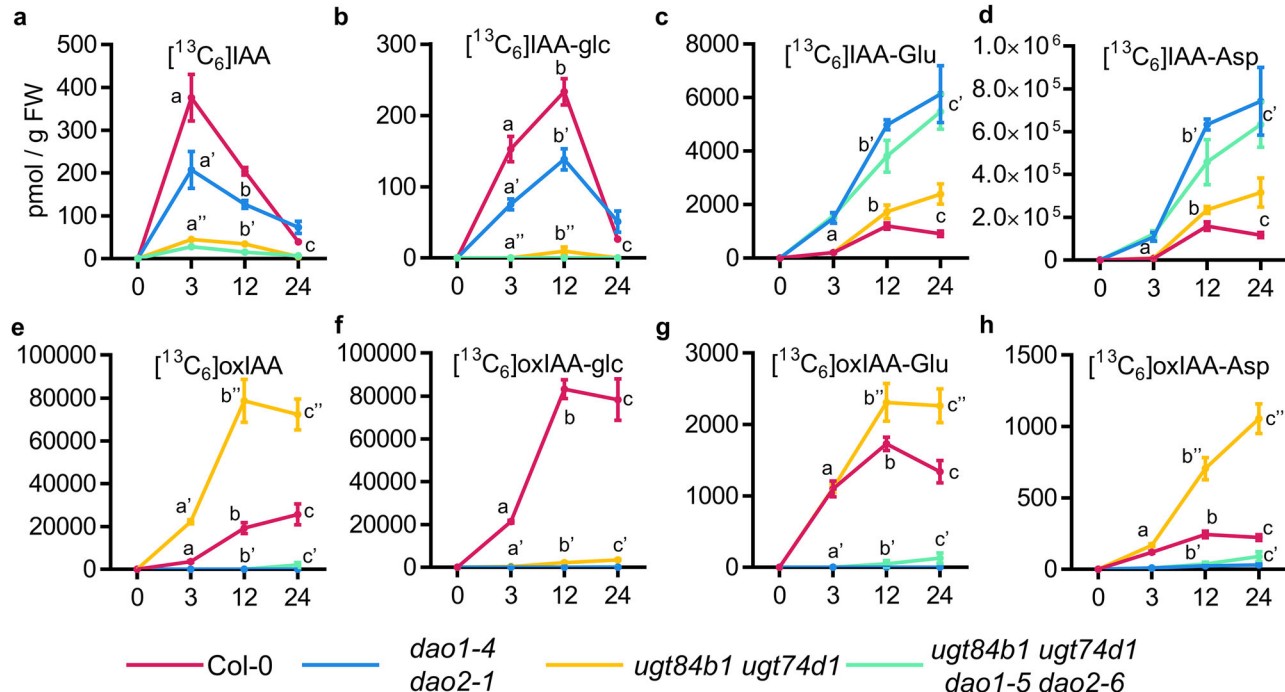

**Fig. 4 | De novo synthesis of indole-3-acetic acid (IAA) metabolites in different Arabidopsis *DAO* and *UGT* multiple mutant lines. a–h** Formation of [$^{13}C_6$]-labelled IAA metabolites in 7-day-old seedlings of the indicated genotypes after incubation with 1 µM [$^{13}C_6$]IAA for 0, 3, 12, and 24 h. For each time point mean ± standard error of the mean is shown. Levels are expressed as picomoles per gram of fresh weight for (**a**) [$^{13}C_6$]IAA, **b** [$^{13}C_6$]IAA-Glc, **c** [$^{13}C_6$]IAA-Glu, **d** [$^{13}C_6$]IAA-Asp, **e** [$^{13}C_6$]oxIAA, **f** [$^{13}C_6$]oxIAA-Glc, **g** [$^{13}C_6$]oxIAA-Glu, and **h** [$^{13}C_6$]oxIAA-Asp. For each time point, differences were evaluated by one-way ANOVA followed by pairwise comparisons using Tukey's HSD test (*N* = 5).

and IAA-Asp (Fig. S3), the feeding experiment revealed a surprising reactivation of the de novo formation of both oxIAA and oxIAA-aa (Fig. S2e, g, h), with the double *dao1 dao2* mutant unable to produce these metabolites under our experimental conditions (Fig. 4e, g, h). Consistently, the accumulation of oxIAA-Asp in the roots of *ugt74d1 dao1 dao2* seedlings was markedly higher than in wild-type or *dao1 dao2* roots (Fig. S2g). Likewise, oxIAA-Glu was detectable in the roots of the *ugt74d1 dao1 dao2* but not in those of the *dao1 dao2* mutant (Fig. S3f).

**Disruption of the GH3s and DAOs enhances flux through the IAA glycosylation pathway**

We next examined the IAA inactivation dynamics in the absence of both the IAA-amino acid conjugation and oxidative routes by performing [$^{13}C_6$]IAA feeding experiments in the wild type (same profile as above), *gh3oct*, *gh3oct dao1-6*, and *gh3oct dao2-7* genotypes (Fig. 5). All three mutant backgrounds exhibited a reduced capacity to metabolize excess IAA, with the effect being more pronounced in *gh3oct dao1-6* plants (Fig. 5a). Concurrently, [$^{13}C_6$]IAA-glc formation was elevated in all three genotypes (Fig. 5b). As expected, *gh3oct* plants were severely impaired in redirecting IAA towards IAA-Glu or IAA-Asp (Fig. 5c, d), and consistent with DAOs acting downstream of GH3s[20,25], the formation of oxIAA and oxIAA-glc was strongly reduced in the *gh3oct* background (Fig. 5e, f). However, the differential formation of [$^{13}C_6$]oxIAA-aa conjugates in *gh3oct* and *gh3oct dao1-6* plants (Fig. 5g, h), along with the strikingly increased accumulation of [$^{13}C_6$]IAA-Glu in the *gh3oct dao1-6* mutant (Fig. 5c) and the presence of endogenous levels of IAA-Glu in *gh3oct* roots (Fig. S3b), suggests that either non-group II GH3s are capable of conjugating IAA under high-hormone conditions, or that a truncated but partially functional GH3 isoform is still produced from one or more of the disrupted loci in the insertional *gh3oct* mutant.

To better understand this phenomenon, we assayed the response of *gh3oct* plants to kakeimide (KKI)[26], a chemical inhibitor of the GH3 activity. We vertically grew Col-0 and *gh3oct* plants for one week in half-strength MS (mock) and in media supplemented with 5 µM of KKI. We then scored the primary root length and lateral root density. As previously reported[19] and further reproduced in this work (Fig. 3a, b), *gh3oct* seedlings exhibited more lateral roots without a reduction in primary root length at 7 days (Fig. S4). While one would expect the *gh3oct* mutant to be fully insensitive to KKI treatment, we found that primary root growth in *gh3oct* is inhibited by KKI to a similar extent as in the wild type. Similarly, root branching was also enhanced upon treatment with KKI. These results suggest that residual GH3 conjugation activity remains in the *gh3oct* mutant and can be fully inhibited by KKI. These new observations help resolve the apparent discrepancy between the phenotypes of the *gh3oct* mutant and those of a similar mutant generated by CRISPR/Cas9 and reported at a similar time[27].

To study the response of the IAA inactivation network to a fully blocked GH3-DAO pathway (Fig. 1), we performed a [$^{13}C_6$]IAA feeding experiments in triple amidohydrolase *ilr1-1 iar3-2 ill2-1* mutant plants in which the IAA-conjugating GH3 activity was chemically inhibited with kakeimide (KKI[26]; Fig. 6). Using the same condition as in ref. 25 (50 µM KKI with 0.2 µM [$^{13}C_6$]IAA), the formation of [$^{13}C_6$]IAA-aa conjugates was abolished in both wild-type and *ilr1-1 iar3-2 ill2-1* plants (Fig. 6c, d). While [$^{13}C_6$]oxIAA-aa conjugates were only marginally detected in the wild-type, they significantly accumulated in the *ilr1-1 iar3-2 ill2-1* mutant (Fig. 6g, h). The increased levels of [$^{13}C_6$]IAA-glc in GH3-inhibited *ilr1-1 iar3-2 ill2-1* plants indicate a redirection of excess IAA towards the IAA glycosylation pathway (Fig. 6b). Remarkably, no [$^{13}C_6$]oxIAA and [$^{13}C_6$]oxIAA-glc were produced in *ilr1-1 iar3-2 ill2-1* plants (Fig. 6e, f), supporting that DAOs act exclusively in the GH3s pathway[25].

**Transcriptomics supports the auxin hyperaccumulation phenotypes in *gh3oct dao1* and implicates non-group II GH3s in the response to high IAA**

We then decided to carry out a transcriptomic approach to identify the molecular pathways underlying the phenotypes and the unique responses to exogenous IAA in *gh3oct* and *gh3oct dao1-6* mutant lines. To avoid comparing organs, we used 5-day-old seedlings, as these genotypes exhibited

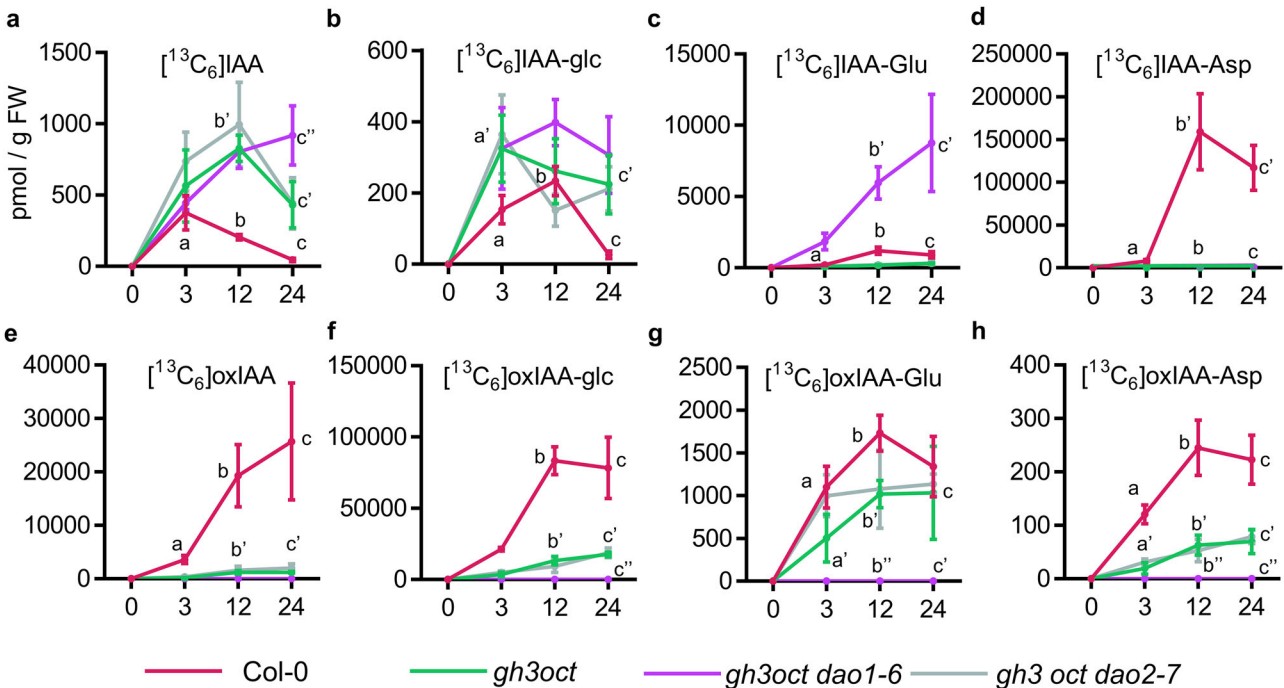

**Fig. 5 | De novo synthesis of indole-3-acetic acid (IAA) metabolites in different *Arabidopsis GH3* and *DAO* multiple mutant lines. a–h** Formation of [$^{13}C_6$]-labelled IAA metabolites in 7-day-old seedlings of the indicated genotypes after incubation with 1 µM [$^{13}C_6$]IAA for 0, 3, 12, and 24 h. For each time point mean ± standard error of the mean is shown. **a–h** Levels are expressed as picomoles per gram of fresh weight for **a** [$^{13}C_6$]IAA, **b** [$^{13}C_6$]IAA-Glc, **c** [$^{13}C_6$]IAA-Glu, **d** [$^{13}C_6$]IAA-Asp, **e** [$^{13}C_6$]oxIAA, **f** [$^{13}C_6$]oxIAA-Glc, **g** [$^{13}C_6$]oxIAA-Glu, and **h** [$^{13}C_6$]oxIAA-Asp. For each time point, differences were evaluated by one-way ANOVA followed by pairwise comparisons using Tukey's HSD test ($N = 5$).

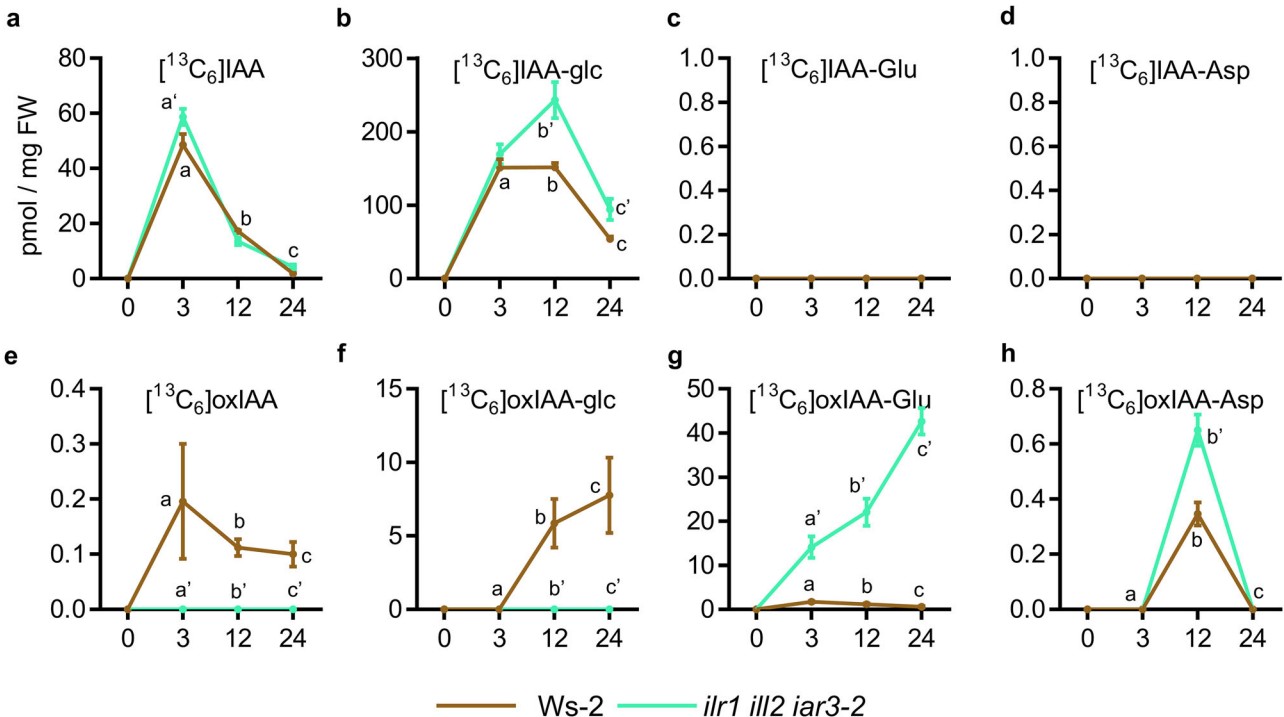

**Fig. 6 | De novo synthesis of indole-3-acetic acid (IAA) metabolites in wild-type (Ws-2) and triple hydrolase *ilr1-1 iar3-2 ill2-1* mutant plants co-treated with KKI. a–h** Formation of [$^{13}C_6$]-labelled IAA metabolites in 7-day-old seedlings of the indicated genotypes after incubation in liquid medium with 50 µM KKI for 12 h, followed by 0.2 µM [$^{13}C_6$]IAA for 0, 3, 12, and 24 h. For each time point mean ± standard error of the mean is shown. **a–h** Levels are expressed as picomoles per gram of fresh weight for **a** [$^{13}C_6$]IAA, **b** [$^{13}C_6$]IAA-Glc, **c** [$^{13}C_6$]IAA-Glu, **d** [$^{13}C_6$]IAA-Asp, **e** [$^{13}C_6$]oxIAA, **f** [$^{13}C_6$]oxIAA-Glc, **g** [$^{13}C_6$]oxIAA-Glu, and **h** [$^{13}C_6$]oxIAA-Asp. For each time point, differences were evaluated by one-way ANOVA followed by pairwise comparisons using Tukey's HSD test ($N = 5$).

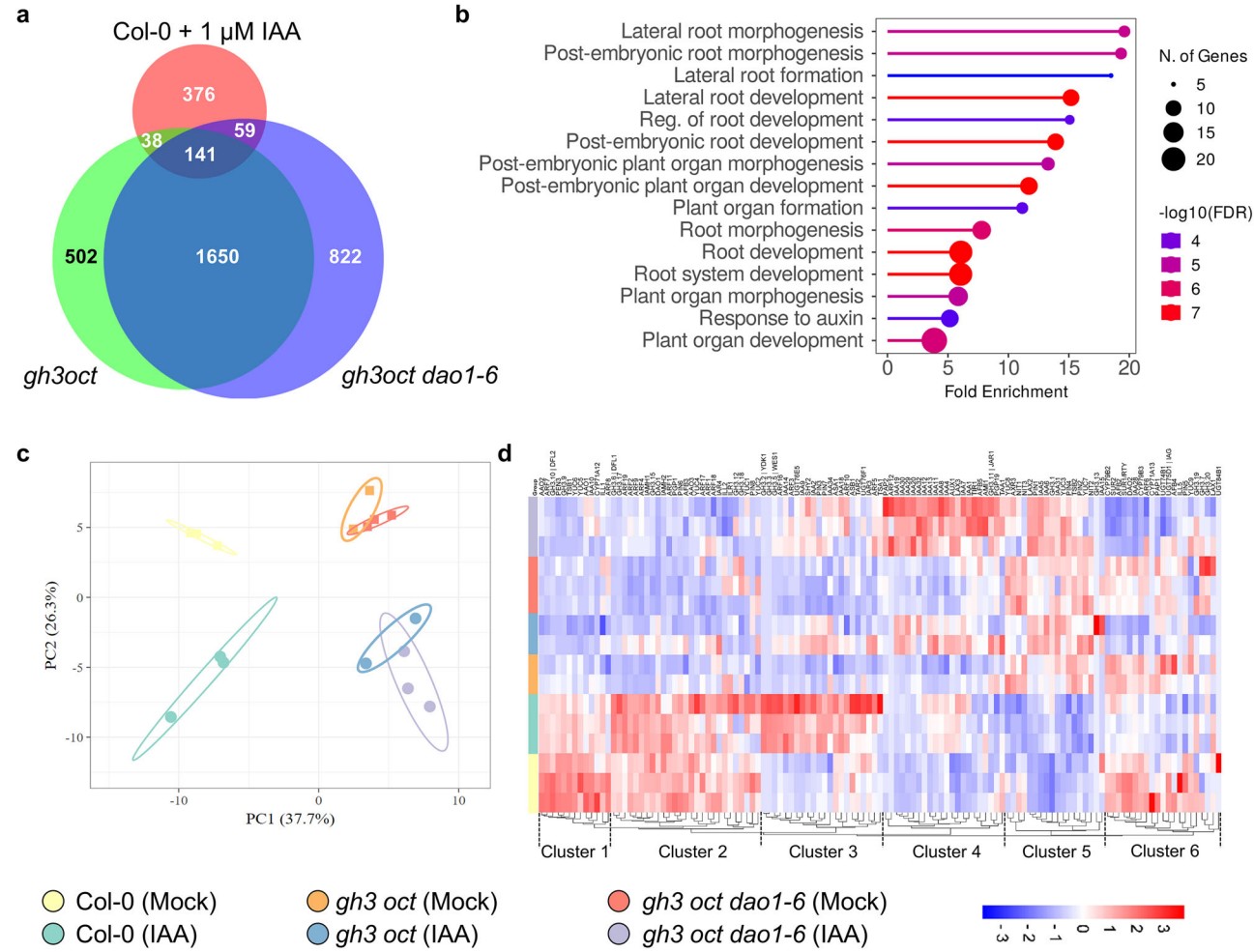

**Fig. 7 | Transcriptomic analyses support that IAA excess causes pleiotropic defects in *gh3oct* and *gh3oct dao1-6*. a** Venn diagram showing the overlap between the response of IAA-treated Col-0 plants and mock-treated *gh3oct* and *gh3oct dao1-6* plants. **b** Lollipop plot showing overrepresented Gene Ontology terms in the Biological Process category for overlapping auxin-related differentially expressed genes (DEGs) identified in IAA-treated Col-0 plants and mock-treated *gh3oct* and *gh3oct dao1-6* plants. The colour scale corresponds to the −log10 of the false discovery rate (FDR); the size of each lollipop represents the number of genes in each category; and the x-axis indicates fold enrichment. **c**) Principal component analysis of transcriptional responses across genotypes and treatments, with samples coloured by group. Ovals indicate clustering of samples. **d** Hierarchical clustering of transcriptional responses across samples. The colour scale indicates the range of normalised log2 fold change values.

similar plant architecture at this stage. To better understand how plants lacking a functional GH3-DAO pathway respond to increased IAA, we also included a set of samples treated with 1 μM IAA for 4 h. The effectiveness of the IAA treatment was first confirmed by qPCR-based transcriptional induction of known IAA-responsive genes in this setup (Fig. S5a), and later verified later using RNA-seq reads on the same genes (Fig. S5b).

We first investigated the eight group II *GH3* genes to try to understand the potential remaining functionality in one or more of them in the *gh3oct* backgrounds. The *gh3oct* combines 8 insertions (either T-DNAs or other forms of similar transference DNA) that can generate different changes in the gene functionality. The first one is just interrupting the natural transcription of the gene. We observed this at *GH3.1* (Fig. S6a), where there seems to be transcription in both sides of the insertion although there are no wild-type transcripts. Actually, by standard RNA-seq analysis (i.e. counting reads), it seems that *GH3.1* is even upregulated in the *gh3oct dao1* because the gene has not lost their IAA-responsiveness. This observation, on the other hand, further supports that *gh3oct dao1* accumulates more IAA than *gh3oct*.

Another usual scenario for chromatin regions with T-DNAs is the silencing of the surrounding environment. Frequently, the small RNA machinery identifies T-DNA insertions as potentially harmful sequences

that have to be heterochromatinized. This is what is seen at different extents in *GH3.2*, *GH3.3*, *GH3.6*, and *GH3.9* (Fig. S6b, c, f, g). Either before or after insertion, there appears to be a complete lack of transcription, likely driven by heterochromatinization of the region. In the tissue used for this experiment, *GH3.4* is not expressed under any condition or genotype (Fig. S6d).

However, we found two insertions that are leaking some functional IAA conjugation activity because they are at the very end of the genes: *GH3.5* and *GH3.17*. While both insertions have generated a partial silencing effect, as evidenced by the signal intensity without and especially with IAA (Fig. S6e, h), we cannot rule out that, at some frequency, some of these late-truncated transcripts generate at least partially functional peptides with some GH3 activity. Nevertheless, considering the different phenotypes and IAA feeding responses, we decided to continue with the transcriptomics analyses.

In absolute terms, the biggest difference in the number of differentially expressed genes (DEGs) was observed between Col-0 and *gh3oct* or *gh3oct dao1*, with more than 2000 DEGs (Figure S7). As a reference, the IAA treatment triggered the deregulation of about 600 genes either in the wild type (Figs. 7A; S7) or in the *gh3oct* and *gh3oct dao1* mutants (Fig. S7). Commonly upregulated genes between *gh3oct* and *gh3oct dao1* in mock conditions were enriched in GO terms such as root meristem growth and

regulation of root development of lateral root development, already pinpointing a precocious activation of lateral root developmental programs (Fig. S8). Genes found to be upregulated only in *gh3oct* but not in *gh3oct dao1* were enriched in cell wall organization processes, which may also be related to lateral root organogenesis. Finally, genes found upregulated only in *gh3oct dao1* plants were highly enriched in translational-related processes (Fig. S8). Downregulated genes found in both mutants were enriched for a myriad of terms related to processes such as cytoskeleton, RNA metabolism, or cell cycle (Fig. S9). Particularly in *gh3oct dao1*, downregulated genes were enriched in cell division and RNA metabolism terms, while no Biological Process term was enriched among the genes found downregulated only in *gh3oct* (Fig. S9).

Analyses of the IAA response revealed a similar number of DEGs across all the genotypes (Figs. S10, S11). GO terms were, as expected, related to auxin responses, including callus formation, wound healing, or lateral root morphogenesis (Fig. S10). Genes only upregulated in one or both multiple mutants were also mainly associated with hormonal responses, including auxin and abscisic acid (Fig. S10). Several genes were upregulated in Col-0 upon IAA treatment, with no response in the mutant background and were enriched in GO terms related to catabolic pathways of isoprenoids and lipids (Fig. S10). Terms related to cell-cell organization were enriched among the commonly downregulated genes in response to IAA across all genotypes (Fig. S11), while two specific responses were observed in *gh3oct* mutants and Col-0. In the mutants, the glucosinolate pathway appears to be downregulated, whereas in the wild type, the root hair development pathway is repressed by IAA treatment (Fig. S11).

In line with our metabolic and phenotypic data, 141 DEGs were shared by IAA-treated Col-0 and mock *gh3oct* or *gh3oct dao1* plants (Fig. 7a). Gene ontology analysis of these common DEGs revealed substantial enrichment for genes involved in lateral root formation, root development, or auxin response (Fig. 7b). With this transcriptomic line of evidence further supporting the role of auxin accumulation underlying the mutant´s phenotypes, we took a closer look at a custom matrix of 137 auxin-related genes in the transcriptomic samples (Supplementary Data 1a). A principal component analysis explained 64% of the variability among samples and identified different patterns of transcriptional misregulation, with four main clusters: (1) Col-0 mock, (2) IAA-treated Col-0, (3) both *gh3oct* and *gh3oct dao1* mock plants, and (4) IAA-treated *gh3oct* and *gh3oct dao1* genotypes (Fig. 7c). We then performed hierarchical clustering of the IAA-related gene matrix for these samples, defining seven clusters (Fig. 7d; Supplementary Data 1b). Clusters 2 and 3 contain genes that, starting from different expression levels in the wild type, were induced upon auxin treatment, with lower induction in the mutants. Interestingly, cluster 4 grouped genes that showed mild, if any, upregulation in Col-0 upon IAA treatment, but whose induction was high in *gh3oct* and even higher *gh3oct dao1*. This enhanced induction is very likely due to the mutant plants´ insufficient ability to inactivate IAA. This cluster included the auxin response factor *ARF12*; auxin transporters like *AUX1*, *LAX3*, and *PGP19*; auxin receptors such as *TIR1*, thirteen members of the *IAA* (*INDOLE-3-ACETIC ACID INDUCIBLE*) family, including *IAA1* (also known as *AUXIN RESISTANT 5*; *AXR5*), *IAA4* (*AUXIN INDUCIBLE 2-11*), *IAA7* (*AXR2*), *IAA8*, *IAA11*, *IAA12* (*BODENLOS*), *IAA13*, *IAA19* (*MASSUGU 2*), *IAA20*, *IAA27* (*PAP2*), *IAA29*, *IAA30*, and *IAA32* (*MEE10*) and *AFB5*, and although counterintuitive, aligned with previous modelling work[28], the auxin biosynthesis genes *AMI1* and *TAA1* (Supplementary Data 1b). Using the AGRIS tool, we further analysed the promoters of the different clusters, looking for potential enrichment in genes carrying auxin-response *cis* elements. Since this custom gene matrix is related to auxin processes, there was a general enrichment of the auxin response element. Among the clusters, clusters 3 and 4 were found to have auxin response *cis* elements in more than 77% of the genes. This observation further reinforces the idea that the genes in the cluster 4 are differentially upregulated in the mutants due to IAA excess (Supplementary Data 1c).

Interestingly, cluster 4 also included *GH3.11* (*JAR1*), a non-group II GH3 member which has been shown to conjugate isoleucine to jasmonic acid (JA)[29]. This observation opened up the possibility that other non-group II GH3 members may also participate in the response to excess IAA in the absence of group II *GH3* and *DAO1*. We specifically examined the 12 non-group II *GH3* expression levels, finding a trend showing *GH3.11* (although not significant in two-way ANOVA), *GH3.13*, and *GH3.14* (At5g13360) with an enhanced response to IAA exclusively in the *gh3oct dao1* background. *GH3.15*, whose role in indole-butyric acid conjugation has been reported[30], showed a higher expression in response to IAA in the *gh3oct* mutant (Fig. S12). Notably, recombinant group III GH3 isoforms, mainly GH3.12, and to a minor extent GH3.15 and GH3.14, exhibited IAA-conjugating activity with Glu[21], suggesting that the residual formation of IAA-Glu observed in the *gh3oct* mutant[19] (Figs. 5c, S3b) could be ascribed to the activity of these non-group II GH3 members rather than the limited and truncated transcripts of *GH3.5* or *GH3.17*.

Another conspicuous trait of the *gh3oct* mutant is a strong early flowering phenotype[19]. Using a curated list of genes involved in flowering time regulation[31] [Supplementary Data 2], we analysed the effect of the *gh3oct* and *gh3oct dao1* mutations on the transcriptional output of this cohort of genes. Among the clearest candidates to explain this trait, we found a strong downregulation of the MADS-box transcription factors *FLOWERING LOCUS C*[32], *MADS AFFECTING FLOWERING4* (*MAF4*), and *MAF5*[33] or *TERMINAL FLOWER1*[34], as well as derepression of *GAox1*, *GA20ox1*[35], *AGAMOUS-LIKE14*[36], or *REPRESSOR OF UV-B PHOTO-MORPHOGENESIS 2*[37] (Fig. S13). Whether there is a direct mechanistic connection between IAA accumulation and the flowering induction remains to be determined.

## Discussion

Since the identification of inactive IAA metabolites by earlier studies[12,38], our understanding of the role of these metabolites and the pathways involved has greatly evolved, from being considered as merely static storage or waste products to the notion that IAA inactivation routes are active and regulated processes, crucial for maintaining auxin levels and spatiotemporal distribution in plants. In this study, we systematically dissected the contribution of three major inactivation routes for IAA metabolic regulation in *Arabidopsis thaliana*: oxidation via DAO enzymes, conjugation to amino acids by GH3s, and glycosylation mediated by UGTs (Fig. 1), generating a comprehensive mutant toolkit combining these pathways and assessing their metabolic, developmental, and transcriptomic consequences.

Understanding of DAO-mediated IAA oxidation has undergone a significant conceptual shift since the identification of the DAO enzymes[24]. Initially, IAA was thought to be the substrate of DAOs to produce oxIAA[22–24,39], and thus DAOs and GH3 were considered parallel and redundant pathways for IAA metabolic inactivation. Later studies showed that the product of the GH3s, IAA-amino acid conjugates, are the primary substrates for DAOs[20,25], and further evidenced that GH3s and DAOs function in a single, linear, and major route for IAA inactivation[25]. Our [$^{13}C_6$]IAA feeding experiments, particularly the complete loss of de novo oxIAA and oxIAA-glc formation in the *ilr1 iar3 ill2* background, provide direct genetic evidence that DAO enzymes act exclusively on IAA–amino acid conjugates rather than on free IAA in vivo.

A major contribution of the present work concerns the functional importance of the IAA glycosylation pathway, conventionally overlooked mainly due to the absence of obvious phenotypes in single and doubles *ugt* mutants[14–16,40] (Fig. 3). Thus, IAA glycosylation should not be viewed as a passive storage route, but rather as a dynamically engaged buffering mechanism that limits transient auxin accumulation when the primary GH3–DAO pathway is compromised or saturated. Even though single *ugt* mutants are phenotypically silent, we show here that knocking-out both *UGT74D1* and *UGT84B1*, two functionally redundant UDP-glycosyltransferases that act on the same substrates[14,41], results in noticeable local auxin overproduction, as evidenced by the elongated hypocotyls and the increased formation of IAA-aa conjugates in *ugt74d1 ugt84b1* seedlings. Our findings strongly indicate that IAA glycosylation significantly contributes to plant development as an active regulatory component of

auxin homeostasis, while if this contribution is largely masked, it is due to functional redundancy within the UGT pathway and with other IAA inactivation pathways. The relatively mild phenotypes of *ugt74d1 ugt84b1* mutants under standard growth conditions and the more pronounced defects when in combination with *dao* mutations, however, underscore that IAA glycosylation, while impactful, plays a context-dependent role.

An intriguing aspect of our findings is the elevated IAA levels observed in the *gh3oct dao1* mutant, accompanied by both phenotypic and transcriptional signatures of auxin overaccumulation. Transcriptomic profiling indeed revealed that *gh3oct* and *gh3oct dao1* mutants considerably mimic IAA-treated wild-type plants at the gene expression level. Given that DAOs act downstream of GH3s, the enhanced auxin response in the nonuple *gh3oct dao1* mutant suggests that accumulated IAA-aa conjugates may be hydrolysed back to free IAA. This is counterintuitive, as one would expect these inactive forms to serve as a buffer or sink under high IAA conditions, rather than being mobilized. These observations imply that conjugate hydrolysis mediated by ILR1/ILL amidohydrolases primarily responds to substrate availability rather than cellular auxin status. Notably, blocking the activity of the GH3s in combination with ILR1/ILLs-mediated hydrolysis does not impair the plant´s capacity to remove excess IAA, in contrast to the observed effect when GH3s alone are knocked down. In both scenarios, however, the IAA glycosylation was upregulated. These results imply that when both the formation and breakdown of IAA-aa conjugates are blocked, the metabolic system compensates by redirecting IAA towards other inactivation pathways, likely involving alternative or yet unidentified pathways. Further research might unravel a broader metabolic flexibility in auxin inactivation than currently understood.

Although the *gh3oct* mutant has been previously used as a genetic proxy for loss of group II GH3 activity[19], our new transcriptomics revealed that it should be considered a strong hypomorph rather than a null background. This residual activity likely reflects low-level expression of late-truncated *GH3.5* and *GH3.17* transcripts, potentially explaining previous phenotypic discrepancies between *gh3oct* and similar mutants obtained by editing[27]. Our transcriptomic analysis also identified a subset of auxin-responsive genes that was more strongly induced in the mutants than in IAA-treated wild-type plants, consistent with impaired feedback regulation due to compromised IAA inactivation capacity. Notably, this set included auxin biosynthetic genes, transporters, and numerous IAA-inducible genes, collectively indicating auxin hyperaccumulation. Interestingly, we also observed upregulation of non-group II GH3 members such as GH3.11, GH3.14 and GH3.15, suggesting a potential role for these enzymes in the response to high IAA conditions. While GH3.11 is considered highly specific for the oxylipin JA as an acyl acid substrate, thereby making a direct involvement in IAA conjugation very unlikely[42,43], evidence exists for the group III members GH3.14, GH3.15, and GH3.12 having a relaxed substrate preference[21]. Whether these enzymes contribute to IAA conjugation in vivo under certain conditions will require further demonstration.

Another emerging and still enigmatic field of knowledge is the link between auxin and translation-related processes. Several lines of evidence have shown a connection between these two processes, such as the auxin-related phenotypes found in mutants for ribosomal subunits[44–47], the translational control of some ARF2, ARF3, or ARF6 through their upstream open reading frames[48], and the existing feedback between auxin signalling and the energetic status via the translational control exerted by TOR on the ARFs[49]. Our transcriptomic analysis of the nonuple *gh3oct dao1* mutant revealed a remarkable upregulation of translation-related GO terms, providing yet another piece of evidence for the connection auxin and translation. Additional research will be required to determine whether the accumulation of IAA alters translation generally or specifically in a subset of mRNAs.

Lastly, our transcriptomics analysis of *gh3oct* mutants also sheds light on their early-flowering phenotype. Several observations support a likely indirect role for auxin signalling in the control of flowering time. The double mutant in auxin biosynthetic genes *yuc1 yuc4*, and the polar transport mutant *pin1*, exhibits a delayed flowering transition[50]. Similarly, gain-of-function alleles of *AXR2* showed a delayed flowering time in Arabidopsis[51], and *ARF4* knock-down lines in *Fragaria vesca* displayed a comparable delay[52]. The downregulation of floral repressors such as *FLC* or *MAF4* and *MAF5* in the IAA-accumulating *gh3oct* and *gh3oct dao1* mutants suggests that auxin overaccumulation may promote early flowering through transcriptional modulation of flowering time integrators. Whether this link is direct or indirect will require further investigation.

Collectively, our findings reveal auxin inactivation as a flexible, multi-layered metabolic network capable of redistributing flux across distinct enzymatic routes to preserve hormonal homeostasis. Our work provides a conceptual framework and genetic tools to uncover novel components of the auxin metabolic network and to understand how plants maintain hormonal balance in the face of genetic and environmental perturbations.

## Methods
### Plant material and growth conditions
We cultured plants as in ref. 53. Briefly, seeds from the *Arabidopsis thaliana* (L.) Heynh. Wild types (Col-0 and Ws-2) and mutant lines were surface-sterilized with 40% v/v commercial bleach and 0.002% Triton-X-100) for 10 min and then washed four times with sterile deionized water. Seeds were stratified for a minimum of 2 days and then sowed under sterile conditions on square petri dishes containing half-strength Murashige & Skoog salt mixture (M0221; Duchefa Biocemie, Haarlem, the Netherlands), 0.05% MES hydrate (M2933; Sigma), and 0.8% plant agar (P1001; Duchefa Biochemie) with the pH adjusted to 5.7 with potassium hydroxide. Plants were kept in vitro for a maximum of 2 weeks, after which they were transferred to pots containing a 3:1 mixture of organic soil and vermiculite. All plants were grown under long-day conditions (16 h:8 h, light:dark) at 22 ± 1°C under cool white fluorescent light (150 µmol photons m$^{-2}$ s$^{-1}$). T-DNA insertional lines *dao1-1*[22], *dao2-1*[23], *gh3* octuple mutant[19], *ugt74d1*[15], and *ilr1-1 iar3-2 ill2-1*[54], as well as the CRISPR/Cas9-generated line *ugt84b1*[14] were previously reported.

### CRISPR/Cas9 plasmid construction
The CRISPR/Cas9-based vector to knock-out the *DAO1-DAO2* locus was constructed using the GreenGate system[55] as described in refs. 14,56. Briefly, 4 guide RNAs (sgRNAs) targeting the *DAO1-DAO2* locus (g#7, g#4, g#6, g#3; Fig. 2; Table S3) were designed using CRISPR-P (http://crispr.hzau.edu.cn/cgi-bin/CRISPR2/CRISPR). The gRNAs were generated using the primers listed in Table S1 and cloned into GreenGate D and E modules by digestion-ligation[55]. The mCherry sequence was amplified from the pGGC015 plasmid using the mCherry-*Bas*I primers (Table S1) and cloned into a B module by digestion-ligation[55]. Two supermodules were then generated by assembling the different GreenGate modules into the intermediate plasmid vectors pGGM000 and pGGN000 (Table S3). The M and N supermodules were then combined into the destination vector pGGZ003 to create the final construct (construct #1; Table S3). A separate GreenGate assembly, including 2 additional sgRNAs targeting the *DAO1* coding sequence (g#2, g#5; Fig. 1; Table S3), into the destination vector pGGZ003 was performed to generate construct #2.

Additionally, a CRISPR/Cas9 construct to knock out *DAO1* (construct #3) was generated in the pKI1.1 R plasmid following the protocol described in ref. 57. Briefly, the circular pKI1.1 R plasmid was linearized by incubating 1.5 µg of the purified plasmid with the *Aar*I restriction enzyme for 10 h and then dephosphorylated using FastAP (Thermo Fisher). A target-specific gRNA was designed (g#1) using CRISPR-P 2.0 (http://crispr.hzau.edu.cn/cgi-bin/CRISPR2/CRISPR). Oligonucleotides harbouring the gRNA target (Table S1) were hybridised by slow cooling from 95-25°C and then phosphorylated using T4 polynucleotide kinase (NEB). The digested plasmid and the hybridised oligonucleotides were ligated using T4 Ligase (NEB) and then transformed into *E. coli* DH5alpha competent cells. The correct assembly of GreenGate intermediate and destination vectors was confirmed by restriction analysis. The sequence integrity in all modules and destination

vectors was verified by Sanger sequencing. All constructs were mobilised into *Agrobacterium tumefaciens* GV3101 (C58C1 Rif R) cells by electroporation.

## Plant transformation and isolation of transgene-free edited mutant lines

All constructs were transferred to Arabidopsis plants by floral dipping[58]. $T_1$ transgenic plants were selected on plates supplemented with 15 mg l$^{-1}$ hygromycin B (Invitrogen).

Transformation of Col-0, *ugt74d1*, and *gh3oct* plants with construct #1 allowed us to isolate the CRISPR/Cas9 *dao2*, *ugt74d1-1 dao1 dao2*, and *gh3oct dao2* mutant lines, respectively. The *dao1 dao2* double mutant was isolated after transformation of *dao2-1* plants with the construct #2 plasmid. Transformation of *gh3oct* plants with construct #3 led us to isolate the *gh3oct dao1* mutant. To generate the *ugt74d1 ugt84b1* mutant we transformed *ugt74d1* plants with our previously generated CRISPR/Cas9 construct targeting *UGT84B1*. The construct and the nature of the isolated deletions in *UGT84B1* are as previously described. The remaining genotypes were obtained by crossing and genotyping.

## Morphometric measurements

For root and hypocotyl phenotyping, vertically grown plates were imaged using Epson Perfection V600 photo scanners. Lengths were measured from scaled images using FIJI software[59]. Differences were evaluated by a one-way ANOVA followed by pairwise comparisons using Tukey´s HSD test.

For root phenotyping in response to KKI treatment, Col-0 and *gh3oct* seeds were grown vertically in media supplemented with 0 (mock) or 5 μM KKI under long-day conditions (16 h light/8 h dark) in cultivation chambers maintained at 21 °C, with a light intensity of ~100 μmol m − 2 s − 1 and 60% relative humidity. Root length and lateral root number of more than 20 seedlings per genotype were scored at 7 days after germination. Plates were imaged with a Sony α 7 II camera, and roots were quantified using ImageJ/Fiji. Differences were evaluated by a one-way ANOVA followed by pairwise comparisons using Tukey´s HSD test.

## IAA metabolite profiling

Seven-day-old in vitro grown seedlings (Col-0, *dao1-1*, *dao2-1*, *dao1 dao2*, *ugt84b1 ugt74d1*, *ugt74d1 dao1 dao2*, *ugt84b1 dao1 dao2*, *ugt84b1 ugt74d1 dao1 dao2*, *gh3oct*, *gh3oct dao1*, *gh3oct dao2* mutant lines) were incubated with liquid ½ MS medium containing 1 μM [$^{13}$C$_6$]IAA for 0, 3, 12, and 24 h under gentle shaking and in darkness. Seven-day-old in vitro grown seedlings (Ws-2, *ilr1-1 iar3-2 ill2-1*) were incubated with liquid ½ MS medium and pre-treated with 50 μM KKI for 12 h before feeding with 0.2 μM [$^{13}$C$_6$] IAA and 50 μM KKI for 0, 3, 12, and 24 h under gentle shaking and in darkness. For each time point, 10 mg whole seedlings were collected in five replicates.

Extraction, purification and quantification of targeted compounds (IAA, oxIAA, IAA-Asp, IAA-Glu, oxIAA-Asp, oxIAA-Glu, IAA-glc, and oxIAA-glc) were performed according to ref. 60. Briefly, samples were extracted in 1 mL of cold 50 mmol/L phosphate buffer (pH 7.0) containing 0.1% sodium diethyldithiocarbamate and a mixture of isotope-labelled internal standards. A 200 μL portion of the extract was acidified to pH 2.7 with HCl and purified using in-tip micro solid phase extraction (in-tip μSPE). Eluted samples were evaporated under reduced pressure, reconstituted in 10% aqueous methanol, and analysed using a 1290 Infinity LC system (Agilent Technologies, CA, USA) equipped with a Kinetex C18 column (50 mm × 2.1 mm, 1.7 μm; Phenomenex) and a coupled 6490 Triple Quadrupole MS system (Agilent Technologies, CA, USA).

## RT-qPCR

To test the IAA-treatment setup employed for the RNA-seq, RNA was isolated using the Total RNA Purification Kit (Norgen, Thorold, ON, Canada). DNA was removed using the RNase-Free DNase I Kit (Norgen). First-strand cDNA synthesis was performed with the iScript cDNA

Synthesis Kit (Bio-Rad). The *ACTIN2* gene was used as an internal control for relative expression quantification. Four biological replicates (each being a pool of several plants) were analysed in triplicate. Quantitative PCR (qPCR) reactions were performed in 10 μl reactions containing 4 μl of LightCycler 480 SYBR Green I Master (Roche), 4 μl of PCR-grade water (Roche), 1 μl of the corresponding primer pair (10 μM each), and 1 μl of the cDNA template. The primers used are listed in Table S1. Quantification of relative gene expression was performed using the comparative $C_T$ method ($2^{-\Delta\Delta Ct}$)[61] on a CFX384 Touch Real-Time PCR Detection System (Bio-Rad).

## RNA-seq

For RNA-seq, seedlings of Col-0, *gh3oct*, and the *gh3oct dao1-6* mutant were vertically grown on standard media in square petri dishes on top of a nylon mesh for five days. The nylon meshes were then transferred to petri dishes containing standard media (mock) or standard media supplemented with 1 μM IAA for 4 h. Following treatment, tissue was harvested and immediately flash-frozen. Total RNA was isolated using the RNeasy Plant Mini Kit (Qiagen), and RNA quality was verified by capillary electrophoresis using an Agilent 2100 Bioanalyzer. RNA-seq libraries and the downstream bioinformatics analyses were done by BGI using the MGIEasy RNA Library Prep Set, and sequencing was performed on a DNBSEQ-T7 platform using 100 bp paired-end (PE100) reads at BGI, Hong Kong. Sequencing statistics are shown in Table S2. Clean reads were aligned to the Arabidopsis thaliana TAIR10 reference genome[62] using HISAT2 (v2.2.1)[63], and gene expression was quantified using HTSeq-count[64]. Differentially expressed genes (DEGs) among samples were identified using DESeq2[65], applying a threshold of p-adjust ≤ 0.05 with no fold-change threshold applied. Genotypes were analyzed in triplicate, except for the *gh3oct* mock and IAA treatments, which were sequenced in duplicate because one library failed quality control. Raw sequence data were deposited in the Short Read Archive (SRA) under the BioProject reference PRJNA666323.

## Statistics and reproducibility

One-way ANOVA was used for Figs. 3–6 and S2–S3, followed by Tukey's HSD post hoc tests. Figure 3 shows individual data points together with the mean ± standard error of the mean (SEM). Figures 4–6 are presented as time-course experiments, where each point represents the mean value for each time point ± SEM. Two-way ANOVA followed by Tukey's HSD post hoc test was used for Figs. S4, S12, and S13. For targeted expression analyses, a Student's t-test (Figure S1) and a Mann–Whitney *U* test (Figure S5) were applied as indicated in the corresponding figure legends.

Statistical analyses were performed using GraphPad Prism version 6.0.1. For Fig. 3, the sample size is indicated above each genotype. For Figs. 4–6, five biological replicates (*n* = 5) were analysed per condition. For RNA-seq experiments, all genotypes and treatments were analysed in triplicate (*n* = 3), except for *gh3oct*, which was analysed in duplicate (*n* = 2) due to the loss of one replicate per treatment during library preparation. Sample sizes were determined based on previous experiments.

## Bioinformatics analyses

The sgRNAs used for the CRISPR/Cas9-editing were designed using CRISPR-P (http://crispr.hzau.edu.cn/cgi-bin/CRISPR2/CRISPR). Charts shown in Figs. 2–5, S1–3, S9-S10 were created using GraphPad Prism 6. All the Venn Diagrams were generated using BioVenn[66]. The PCA was created with ClustaVis[67]. The Hierarchical clustering was performed with SRplot[68]. The GO analyses and lollipop charts were built using ShinyGO[69].

## Accession numbers

*DAO1* (AT1G14130), *DAO2* (AT1G14120), *GH3.1* (AT2G14960), *GH3.2* (AT4G37390), *GH3.3* (AT2G23170), *GH3.4* (AT1G59500), *GH3.5* (AT4G27260), *GH3.6* (AT5G54510), *GH3.9* (AT2G47750), *GH3.14* (AT5G13360), *GH3.15* (AT5G13370), *GH3.17* (AT1G28130), *ILR1* (AT3G02875), *ILL2* (AT5G56660), *IAR3* (AT1G51760) *UGT74D1* (AT2G31750), *UGT76E5* (At3g46720), *UGT84B1* (AT2G23260).

**Reporting summary**

Further information on research design is available in the Nature Portfolio Reporting Summary linked to this article.

## Data availability

All data generated or analysed during this study are provided in this published article and its supplementary data files or will be provided upon reasonable request. RNA-seq raw data were deposited in the Short Read Archive (SRA) under the study reference PRJNA666323. Source data is available in Supplementary Data 3.

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

## Acknowledgements

We thank Roger Granbom (UPSC, Umeå, Sweden), and Hana Svobodová and Kateřina Perničková (LGR Olomouc, Czech Republic) for their technical assistance. We also acknowledge the Swedish Metabolomics Centre (https://www.swedishmetabolomicscentre.se/) for the access to the instrumentation.

## Author contributions

R.C.-S., K.L. and E.M.-B designed the methodology. R.C.-S. and E.M.-B. obtained the multiple mutants and performed the phenotypic analyses. AŽ provided materials. R.C-S., F.B., P.H. and A.A. carried out the feeding experiments. A.P., J.Š., and O.N. performed the metabolic profiling. R.C.-S. and E.M.-B. generated the RNA-seq material. E.M.-B. analysed the RNA-seq. E.M.-B, R.C.-S., U.V., M.B., O.N. and K.L analysed the data. E.M.-B., O.N., and K.L. obtained funding and provided resources. R.C.-S. and E.M.-B. prepared the original draft. All authors reviewed and edited the manuscript.

## Funding

This work was partially supported by the Wallenberg Initiatives in Forest Research (WIFORCE) funded by the Knut and Alice Wallenberg Foundation (KAW 2020.0240) as well as the Swedish Governmental Agency for Innovation Systems (Vinnova); the Swedish Research Council (VR); Kempestiftelserna (JCK-1811, JCK-1111); the University of Nottingham (Nottingham Research Fellowship to U.V.) and Royal Society Research Grants (RGS_R1_191323 to U.V.); Unravelling Spatio-temporal Auxin Intracellular Redistribution for Morphogenesis (STARMORPH; 101166880 to A.P., P.H. and O.N.); TowArds Next GENeration Crops (TANGENC, no. CZ.02.01.01/00/22_008/0004581 to A.P., P.H., A.Z. and O.N.); and Grants RYC2021-030895-I (to E.M.-B.) funded by MICIU/AEI/10.13039/501100011033 and by the European Union NextGenerationEU/PRTR and PID2023-147737NA-I00 (to E.M.-B.) funded by MICIU/AEI/10.13039/501100011033 and "ERDF/EU". Open access funding provided by Swedish University of Agricultural Sciences.

## Competing interests

The authors declare no competing interests.
