## [Transparent Peer Review file · Communications Biology]

Comprehensive characterisation of IAA inactivation pathways reveals the impact of glycosylation on auxin metabolism and plant development in Arabidopsis

Corresponding Author: Dr Eduardo Mateo-Bonmati

Version 0:

Reviewer comments:

Reviewer #1

(Remarks to the Author)

In this study, the authors generated a comprehensive set of combinatorial mutants of IAA-inactivating enzymes, including GH3s, DAOs, and UGTs, and then analyzed their resulting phenotypes. The de novo synthesis and steady-state levels of IAA metabolites in these mutants were measured using state-of-the-art LC-MS/MS analysis with isotope-labeled internal standards. This study extensively investigates the metabolic profiles of IAA metabolites at both de novo synthesis and steady-state levels. Finally author performed RNA-seq analyses among gh3oct, gh3oct dao1 and WT+IAA plants.

However, I have several concerns on the genetic materials and statistical analyses used in this study, which are detailed below. These concerns cast serious doubt on the conclusions of this manuscript.

Major Concerns

1. The dao2-1 allele

(Line 140) The authors used the dao2-1 allele to generate the dao1 dao2 double mutant. Zhang et al. (PNAS, 2016; PMID: 27651492) previously reported that this allele contains a T-DNA insertion in the second intron of the DAO2 gene and that dao2-1 plants still express DAO2 at approximately 39% of wild-type levels.

This raises a critical question: Is dao2-1 a true loss-of-function mutant? The residual gene expression suggests it may not be a knockdown allele. Have the authors confirmed the DAO2 expression level in their dao1 dao2 double mutants to ensure it is a functional knockout? This is crucial for the interpretation of the phenotypic and metabolite analyses of dao2-1, and dao1 dao2 mutants

2. Statistical Analysis of Phenotypic Data

(Lines 170, 178; Figure 2) The authors analyzed the phenotypic data in Figures 2B-2D using a Student's t-test to determine if differences between two groups are statistically significant. This is an inappropriate statistical method for an experimental design that involves comparisons across multiple groups.

The authors should re-analyze this data using an Analysis of Variance (ANOVA) followed by a suitable post-hoc test for multiple comparisons, such as Tukey's HSD or Dunnett's Test. Consequently, the conclusions drawn in lines 170 and 178 regarding the hypocotyl lengths of dao2-1 and the ugt84b1 ugt74d1 double mutant must be carefully reconsidered in light of a correct statistical analysis.

Furthermore, the quantitative data of metabolites presented in Figures 3-5, S1, and S2 appear to lack any statistical analysis. All quantitative data throughout the manuscript must be analyzed with appropriate statistical methods, and the results of these analyses should be clearly reported in the table, figures, supplementary table or text.

3. The gh3oct mutant

(Lines 285-291) The gh3 oct mutant used in this study was previously generated by the same first author (Casanova-Sáez et

al., *New Phytologist*, 2022). This paper have demonstrated that IAA-Glu levels were not reduced in the gh3 oct mutant. Consistent with the previous paper, the current manuscript shows that IAA-Glu accumulates in the gh3oct dao1 mutant (Figure 4C, S2B), while IAA-Asp does not.

The T-DNA insertion in the gh3-17 allele (SALK_050597), a component of the gh3oct mutant, is located at the C-terminus of the last exon. This position raises the possibility that a truncated, partially functional protein is produced. This would mean the gh3oct mutant is not a null (complete loss-of-function) mutant for Group II GH3s.

This concern is supported by another study (Guo et al., *BBRC*, 2022), which generated a gh3oct mutant using CRISPR/Cas9. Their mutant exhibited much more severe high-auxin phenotypes (e.g., strong root growth inhibition, more lateral and adventitious roots) than the one used in this work. The phenotypes reported by Guo et al. are also consistent with those induced by chemical GH3 inhibitors (Xie Y, *PNAS* 2022: PMID: 36454752, Fukui K, *PNAS* 2022: PMID: 35914172). This leads to a key question: Is the gh3oct mutant used in this study resistant to the GH3 inhibitor kakeimide (GH3 inhibitor used in this study)?

If the gh3oct mutant is indeed not a null mutant, all claims regarding the function of Group II GH3s based on this genetic background should be carefully revised and toned down. The authors acknowledge this possibility in lines 285-291, but the potential impact of this issue on the paper's conclusions needs to be more thoroughly addressed throughout the manuscript.

4. Metabolite profiling in the ugt74d1 dao1 dao2 mutant

(Lines 231-238) The metabolic profile of oxIAA and oxIAA-amino acid conjugates in the ugt74d1 dao1 dao2 mutant seems to be very surprising. The authors suggest this points to "possible involvement of additional IAA oxidase activities" and a "complexity and flexibility of the IAA inactivation network higher than currently understood."

However, the data in Figures 3, S1, and S2 and previous papers suggest that DAO1 is responsible for almost all oxidation of IAA-amino acid conjugates. The presence of these metabolites in a triple mutant background is therefore unexpected. Have the authors confirmed that the dao1 and dao2 mutations are homozygous in their ugt74d1 dao1 dao2 line? Genotyping confirmation is essential to rule out the possibility of genetic segregation or contamination.

Specific points

in Fig1A upper box.
dao1 dao2 > dao1 ?

Figure S2.

IAA-Asp level in ugt74b1 dao1 dao2 is missing,

Reviewer #2

(Remarks to the Author)

As sessile organisms, plants rely on an extensive set of regulatory molecules among which phytohormones are a crucial to adapt their life cycle in response to changes in the environment. Auxin, whose major active form is indole-3-acetic acid (IAA), is a phytohormone involved in nearly every aspect of plants' life, from embryo development to senescence. Therefore, proper spatiotemporal patterns of auxin biosynthesis, transport, catabolism and response are essential for plants' phenotypic plasticity and fertility. Recently, IAA catabolism through the oxidative pathway has been revised placing GH3 and DAOs activity in the same pathway. However, IAA inactivation through glycosylation, catalyzed by UDP-glycosyltransferases, is another major IAA inactivation pathway. In this study, the authors endeavor to examine the interaction between these pathways compared to their individual roles, which is a major contribution in the auxin field.

In general, the paper is logically structured and reads well. I appreciated the conciseness of the introduction and the technical thoroughness of the metabolic side of the study. While reading the manuscript I was both excited—for the beauty of results presented in it and how the authors described them—and overwhelmed by the magnitude of the study. The authors did a great job driving the reader through the manuscript to highlight the main findings, but it is still not a quick read. I do not have a definitive solution for it, but I wanted to acknowledge the challenge and make some suggestions that I think may facilitate data interpretation.

General commentaries:

The authors introduce here for the first time a set of newly generated lines with altered auxin inactivation and the measurement of relevant morphometric traits strongly regulated by auxin concentration, such as hypocotyl and root length and lateral root density. However, such measurements were not accompanied by representative pictures of plants. While the analyses performed here relied on the abovementioned traits as a readout of the metabolic differences observed between genotypes, some of these lines are introduced for the first time, so it would be helpful to count on such pictures to visualize other traits associated to changes in auxin concentration described for lines with altered auxin homeostasis, like cotyledon epinasty, root branching, or adventitious

root formation, under the growth conditions (and developmental stage) chosen to carry out the metabolic quantifications.

The statistical analyses were in many cases inadequate for the interpretation of the results that are later discussed. In most cases, pairwise comparisons between a mutant and WT were tested but later discussion covering the significance of differences in some traits between mutants is mentioned without having assessed statistical significance (ANOVA).

I found a number of typos, some of which I point out below, but the authors may want to read through the manuscript again to fix them.

Specific commentaries:

Line 53: DAOs are not spelled out as DIOXYGENASE FOR AUXIN OXIDATION for the first time until line 97. However, GH3, UGTs, and IAA are fully spelled out in the abstract.

Line 83: Are both UDP-glycosyltransferases and UDP-glycosyl transferases (line 55) correct?

Line 85: Since it is mentioned that only three UGTs are associated to IAA inactivation in Arabidopsis, and Figure 1 is titled “main pathways for indole-3-acetic acid (IAA) metabolic inactivation in Arabidopsis”, it would be helpful to also place UGT76E5 in the figure.

Line 97: Following the same rationale, I suggest including DAO2 in Figure 1, specially because *dao2* is one of the stacked mutants.

Line 113: Figure 1 title is bold but, through the manuscript, other figures only bolded the name.

Line 115: “inactivation define” should be “inactivation defines.”

Line 165-166: It is not explicitly stated whether “mild auxin-associated phenotypes” refers to phenotypes associated to increased or decreased auxin concentration. Cite a reference showing that increased primary root length and lateral root density have been previously associated as such.

Line 167: DAO 1 should be DAO1 (not separated).

Line 168: “...despite DAO1 being the predominant oxidase in root IAA metabolism”: Add reference.

Line 169: If “... DAO2 contributes to regulating hypocotyl elongation” is known from the literature, cite a reference.

Line 170: Was the difference in hypocotyl length observed in *dao2* compared to WT consistently found across experiment repetitions? If so, how would it be explained the “more complex interplay” leading to *dao1 dao2* not showing such difference?

Line 171: *dao1 dao2* should be italicized.

Line 178: I think it would be helpful to first mention that *ugt84b1 ugt74d1* does not show any defects in roots to understand why it is remarkable that *dao1 dao2 ugt74d1 ugt84b1* shows more prominent increase in LRD (but not in length) than *dao1 dao2*.

Lines 187-188: “...nonuple mutants displayed a further reduction in PRL and a greater increase in hypocotyl length...” is not statistically supported since only pairwise comparison to WT has been tested (t-test) rather than multiple comparison (ANOVA).

Lines 189-190: To my understanding, this statement should be accompanied by some evidence showing that *gh3oct* is fully knocked out. Otherwise, some GH3 leaky activity might be further disrupted by *dao* mutants. Gene expression analysis from RNAseq would be helpful here.

Line 206: Typo: “observed ing *dao1-1*.”

Line 210: Typo: “Trunover”

Line 218: Given the large amount of data, and in order to help the reader understand the

author's point, it would be very helpful to reword the sentence in such a way that instead of saying "ugt84b1 ugt74d1 dao1 dao2 mutant had a stronger impact on lateral root density" it specifies whether LRD was higher or lower and compared to what genotype. For example, "ugt84b1 ugt74d1 dao1 dao2 mutant showed greater LRD than dao1 dao2 (or ugt84b1 ugt74d1)". Otherwise, the reader needs to go back to the cited figure to remember and it interrupts the flow. Nonetheless, this statement needs to be statistically supported by an ANOVA.

Line 221: "Notably, our data showed an identical lower accumulation of [13C6]IAA" is not statistically supported. ANOVA is required for that.

Line 318: In the section titled "RNA-seq supports the auxin hyperaccumulation phenotypes in gh3oct dao1 and implicates non-group II GH3s in the response to high IAA" the authors did not examine the expression GH3s and DAOs in the gh3oct, gh3oct dao1, and gh3oct dao2 backgrounds, which would be essential to understand the degree of disruption that each mutant has in the oxidative pathway.

Line 362: The 64% is explained only by the combination of PC1 and PC2 (the PCs with the greatest percentage), but the summatory of all PCs surely adds up to more than that.

Line 373: I find counterintuitive that the auxin biosynthesis-related TAA1 (cluster 4) is induced in response to IAA in the IAA-inactivation-defective gh3oct dao1 mutant. How would the authors explain that?

Line 447-448: The overlap (238 genes) is 40% of WT + IAA genes but only ~10% of gh3oct-harboring mutants. Considering the number of differentially expressed genes in gh3oct-harboring mutants (2000+ genes) and only ~10% overlapping with WT + IAA, I would think that the alteration in their transcriptional programs due to IAA accumulation is not the most prominent factor. Therefore, I find that "largely mimic IAA-treated wildtype plants" is an overstatement.

Line 460: "involving alternative or yet unidentified pathways": in the introduction, the authors mention IAA methylation (meIAA) as an inactivation pathway. Would it be possible to reanalyze their results to determine meIAA concentration? Also, could the authors show the expression of UGTs in gh3oct or gh3oct dao mutants?

Figure commentaries:

Figure 2

I find panel A confusing and difficult to interpret: the boxes that amplify the DNA/aminoacid fragments that have been modified zoom out from a region that is not clear what region exactly is being amplified. In the spirit of simplifying the message, the authors may want to consider separating the sequence information (maybe as a supplementary figure?) from the structural information (in main figure). For example, placing the WT structure of DAO1 (dark grey)/DAO2 (light grey) on top, below it, the dao1, dao2, or dao1 dao2 structures shading in semitransparent black the disrupted parts of the genes in each background. This is just a format suggestion, but the authors may consolidate it otherwise; I just find panel A busy and unclear. Since the scale bar is explicit in panel A, I do not see the need to mention it in the caption.

Panels B-D are clear and easy to read. In the caption, they are cited as phenotypic parameters, but these are variables, not parameters. To avoid confusion, they can be referred to as morphometric traits. At the same time, as a colorblind person, it is hard for me to tell apart the colors. I do not recommend combining different shades of green and orange in the same plot. I usually try to combine the three primary colors (red/magenta, yellow, blue/cyan) with black, white, and grey, in addition to symbols if needed to scale up the number of unique signatures. In panels B-D, I personally don't see the need to color-code the genotypes since they do not add any information and some of them are hardly distinguishable for me: dao1-1 vs ugt74d1 ugt84b1, dao1 dao2 vs gh3oct dao1, or ugt74d1 dao1 dao2 vs gh3oct. I do see the consistency in color usage for specific genotypes across the manuscript. Therefore, if the authors want to stick to color-coding the genotypes (it might be informative for other non-colorblind fellows), I recommend using other colorblind-friendly selection of colors, or adding different shapes (circle, triangle, asterisk) to more "similar" colors.

To my understanding, and according to the statement in lines 144 and 151, followed up by that in line 193, the quantitative nature of Figure 2 is to examine the interaction between the conjugation and oxidation pathways for IAA inactivation. Therefore, pairwise comparison to WT (Student's t-test) is not adequate for this purpose and, instead, one way-ANOVA should be performed. For example: are the differences in hypocotyl length and lateral root density observed in dao1 dao2 significantly different from those observed between ugt74d1 dao1 dao2 or ugt74d1 dao1 dao2? How about the differences in primary root length observed between ugt84b1 dao1 dao2 and ugt74d1 ugt84b1 dao1

dao2?

Finally, the methods section describes that “plants were kept in vitro for a maximum of 2 weeks”. While I think it is ok for the methods section, it is necessary to specify in Figure 2’s caption the seedlings’ age in the reported experiment for the sake of reproducibility.

Figure 3 & Supp. Figure S1

Statistical tests are missing, but it would be possible to visually assess the statistical significance by substituting Standard Error by Confidence Intervals 95% and looking at overlapping bars.

How do the authors explain that dao1 dao2 mutant has intermediate evolution of 13CIAA-glc? Considering the disruption of IAA-aa oxidation, the expectation would be a compensation through the glycosylation pathway, similarly to what happens in ugt84b1 ugt74d1 mutant and the GH3/oxidative pathway.

Figure 4

Line color is hard to distinguish between gh3oct and gh3oct dao2 (consider using black). If WT values are the same as those used in Figure 3, it should be mentioned.

Figure 6

Panel A: Number of genes in Venn’s diagram are too small to read well. I am aware that now that papers are digital we can zoom in, but the smallest font size should be 6 pt, being 8 pt the recommended font size.

Supp. Figure S2

Statistical tests are missing. It is a major drawback that [IAA-glc] is missing. Also, growth conditions to measure metabolite levels in steady-state (5-day old) are different from those used in Figure 3 & Supp. Figure S1 (7-day old). Considering that this study examines IAA deactivation through the interaction of glycosylation and oxidation pathways, and that Figure 3 shows that dao1 dao2 mutant is impaired in IAA glycosylation, would it be possible to quantify IAA metabolites again to successfully detect IAA-glc (ideally, in 7-day old seedlings to compare to the observations from Figure 3 & Supp. Figure S1)?

Supp. Figure S5

Letters are too small and what bubble represents what genotype is not clear.

Reviewer #3

(Remarks to the Author)

The authors present an interesting study of various fates of radiolabelled IAA with higher order mutants in the different metabolic pathways. The authors found some interesting results regarding the role of IAA metabolites in regulation of flowering time via FLC.

There are a few questions about the study that I’m sure the authors can address.

1. The dao2-1 allele was not fully described in the text, so I looked it up. It appears in previous publications that it is a knockdown allele and not a knockout allele. If this is the case, what do the authors think about the double dao1 dao2 mutant presented here in a dao2-1 knockdown background?

2. Line and 207 “Interestingly, ... oxIAA, oxIAA-glc, and oxIAAaa were still present in both single dao1-1 and double dao1 dao2 mutant plants (Figure S2E-G)...” and Line 228. “Surprisingly, ... de novo formation of both oxIAA and oxIAA-aa”. It appears that the dao2-1 knockdown allele has active oxidation activity in dao2-1 mutant. While it is possible that a de novo oxidation activity is present as the authors suggest, what do the authors think about a simpler explanation that the observed oxidative metabolites were from the activity of the knockdown dao1-2 allele?

3. Of the genes upregulated in the RNAseq analysis, which gene have auxin regulatory elements in their promoters? This information would point to a hierarchy of direct vs indirect effects of ectopic auxin or auxin metabolite accumulations of the higher order loss of function mutants and exogenous IAA treatments.

The text should be reviewed for spelling and grammar.

Version 1:

Reviewer comments:

Reviewer #1

(Remarks to the Author)

The authors have done an excellent job addressing the reviewers’ concerns, and the manuscript has been substantially strengthened by the addition of new data and clarifications. However, two specific points remain that, in my view, require

further attention to ensure the accuracy of the work.

(1) Metabolite profiling in the *ugt74d1 dao1 dao2* mutant (Lines 231-238)

Original review comments by reviewer #1.

Metabolite profiling in the *ugt74d1 dao1 dao2* mutant (Lines 231-238) The metabolic profile of oxIAA and oxIAA-amino acid conjugates in the *ugt74d1 dao1 dao2* mutant seems to be very surprising. The authors suggest this points to "possible involvement of additional IAA oxidase activities" and a "complexity and flexibility of the IAA inactivation network higher than currently understood." However, the data in Figures 3, S1, and S2 and previous papers suggest that DAO1 is responsible for almost all oxidation of IAA-amino acid conjugates. The presence of these metabolites in a triple mutant background is therefore unexpected. Have the authors confirmed that the *dao1* and *dao2* mutations are homozygous in their *ugt74d1 dao1 dao2* line? Genotyping confirmation is essential to rule out the possibility of genetic segregation or contamination.

Author response to the comments by reviewer #1.

We agree that the profile shown by the triple mutant was interesting and which is why highlighted in the first version. However, steady-state levels of IAA-Glu, IAA-Asp, oxIAA or oxIAA-glc are consistent with these plants being homozygous for *dao1 dao2* (Figure S3). Nevertheless, we re-checked the zygosity of these mutations and included a new set of primers required to do so in the Supplemental Table S1. It requires further experimental approaches to figure out what exactly is behind this behaviour but we consider that it is beyond the scope of this work.

Second-round review comments by reviewer #1

Revised version [line 239-241]: This further suggests the possible involvement of additional AA oxidase activities, and points to a complexity and flexibility of the IAA inactivation network higher than currently understood."

>While the authors re-checked the zygosity and updated Supplemental Table S1 with the primers used, the results remain puzzling. I am not fully convinced by the differences in oxIAA, oxIAA-aa, and oxIAA-glc profiles between steady-state conditions and feeding experiments in the *ugt74d1 dao1 dao2* mutants. Specifically, the statement in Lines 239-241 should be removed or moved to the discussion section, as the authors themselves admit that explaining this behavior is beyond the scope of the current study. Furthermore, the similar *dao* mutant, *ugt84b1 dao1-5 dao2-2* in Figure S2 exhibits a profile identical to the *dao1-1* mutant, which contradicts the authors' current interpretation in the Results section.

Second-round review comments by reviewer #1

2. GH3 Activity and Citations (Lines 450-454)

Revised version [Line 450-454]: Notably, recombinant group III GH3 isoforms, mainly GH3.12, and GH3.15 and to a minor extent GH3.14, exhibited IAA-conjugating activity with Glu (21), suggesting that the residual formation of IAA-Glu observed in the *gh3oct* mutant (19) (Figure 5c, Figure S3b) could be ascribed to the activity of these non-group II GH3 members rather than the limited and truncated transcripts of GH5 or GH3.17.

>In Lines 453-454, the authors discuss the involvement of non-group II GH3 members in the residual formation of IAA-Glu in the *gh3oct* mutant. However, there are issues with the evidence provided:

1) Reference 21 is inappropriate here, as the IAA-conjugating activities of GH3.12, 14, and 15 were not investigated in that study.

2) Comprehensive enzyme kinetics for GH3.15 (e.g., J. Biol. Chem. 2018, 293:4277-4288) clearly demonstrate that IAA is not a substrate for GH3.15.

The context in lines 450-454 should be removed or moved to the Discussion section. If moved, the authors must provide correct citations that specifically support the IAA-conjugating activity of non-group II GH3s. The current claim that GH3.12, 14, and 15 contribute to IAA-Glu formation is not supported by the cited literature or known biochemical data.

Ref (21) Brunoni, F. et al. Amino acid conjugation of oxIAA is a secondary metabolic regulation involved in auxin homeostasis. *New Phytol* 238, 2264-2270, doi:10.1111/nph.18887 (2023).

Reviewer #2

(Remarks to the Author)

I appreciate the authors' thorough revision of the manuscript and their clear effort to address the concerns raised in my previous review. The manuscript has improved considerably, particularly with respect to data presentation and statistical rigor. However, several important issues remain unresolved and require further attention.

Overall, the remaining concerns are relatively focused and can be addressed by further refining two main aspects of the study: (1) ensuring that the statistical analyses are fully aligned with the experimental designs, and (2) strengthening the characterization of the *ugt74d1 ugt84b1* double mutant.

For the latter, the manuscript would benefit from (a) an expression analysis of UGT74D1 and UGT84B1 in the *ugt74d1 ugt84b1* double mutant—particularly given that a new mutant allele was generated in this work—and (b) inclusion of representative images of this newly generated mutant to facilitate phenotypic interpretation.

Major issues

1. Interpretation of phenotypes in *ugt* mutants and missing supporting data

In line 501, the authors state that “a major contribution of the present work concerns the functional importance of the IAA glycosylation pathway, conventionally overlooked mainly due to the absence of noticeable phenotypes in single *ugt* mutants.” However, representative images of the newly generated *ugt84b1 ugt74d1* double mutant are not shown in the newly provided Supplementary Figure S1, whereas images for *dao1-4 dao2-1* and the higher-order *ugt74d1 ugt84b1 dao1-5 dao2-2* mutant are included.

In Figure 3, the double mutant shows a small but statistically significant increase in hypocotyl length relative to WT, while no differences in root length or lateral root formation are reported. In the absence of additional phenotypic documentation, it would be more accurate to state that both single and double *ugt* mutants lack obvious auxin-related phenotypes, unless the authors provide visual or quantitative evidence for additional traits that were not measured.

In light of the results for *gh3oct*—which, despite undetectable conversion of IAA to IAA-Glu and IAA-Asp, shows normal sensitivity to KKI and only weak auxin-related phenotypes (potentially explained by the newly disclosed leaky expression of some GH3 genes)—it is reasonable to ask whether a similar explanation could apply to the *ugt74d1 ugt84b1* double mutant. Specifically, residual expression of either gene could potentially account for the lack of strong phenotypes despite near-complete loss of IAA-Glc formation.

Given that this double mutant was generated in the present study through creation of a new UGT84B1 allele, an expression analysis documenting the degree of transcriptional disruption is warranted.

2. Inappropriate statistical analysis in Supplementary Figure S4

The statistical analysis used in Supplementary Figure S4 does not match the experimental design. As the experiment follows a Genotype x Treatment structure, a two-way ANOVA should be performed instead of a one-way ANOVA.

Additionally, figure captions should be written to allow independent interpretation. In this context, abbreviations such as “KKI” should be spelled out in full.

3. Statistical analysis in Supplementary Figure S12

Based on the Genotype x Treatment experimental design, the use of a Student’s t-test in Supplementary Figure S12 is not appropriate. A two-way ANOVA should be applied instead.

4. Inconsistent IAA concentrations across experiments

Throughout the manuscript, the authors consistently use an exogenous IAA concentration of 1 μ M (Figures 4, 5, and 7, and presumably Supplementary Figure S2, although this is not stated). However, a fivefold lower concentration (0.2 μ M IAA) is used to assess IAA deactivation dynamics in the *ilr1-1 iar3-2 ill2-1* mutant (Figure 6). I would think that has something to do with a potentially deadly accumulation of IAA in KKI-treated *ilr1-1 iar3-2 ill2-1* mutant or different sensitivity of *Ws-2* background, but this should probably be explicitly explained.

5. Missing statistical analysis in Supplementary Figure S13

No statistical analysis is described for Supplementary Figure S13. Given the experimental design, a two-way ANOVA is expected and should be reported.

Minor issues

1. Consistency in mutant nomenclature

Mutant names should be used consistently throughout the manuscript. For example, the quadruple mutant *ugt74d1 ugt84b1 dao1-5 dao2-2* is referred to as *dao1-5 dao2-2 ugt74d1 ugt84b1* in lines 176–177.

2. Role of MeIAA in compensatory IAA inactivation

In the rebuttal letter, the authors state that experiments measuring MeIAA levels are beyond the scope of the study. I respectfully disagree with this assessment. The Introduction explicitly discusses IAA inactivation via methylation, and in lines 531–532 the authors propose that metabolic compensation may occur through redirection of IAA into alternative inactivation pathways, potentially including unidentified routes.

Despite this framing, the manuscript does not assess whether MeIAA contributes to such compensation. In this context, the rebuttal response reads as somewhat generic, particularly given the manuscript’s stated goal of providing a “comprehensive characterization of IAA inactivation pathways.”

That said, while inclusion of MeIAA measurements would strengthen the study, I recognize the experimental effort required relative to the incremental insight gained, and therefore I do not insist further on this point.

3. Inconsistent annotation of statistical significance

The annotation of statistical significance groups is inconsistent across figures and may be confusing for readers. Standard practice is to annotate each datapoint with its corresponding significance group. In Figures 4–6, however, each group is labeled with a single letter, requiring the reader to infer datapoint membership based on proximity. Unless explicitly stated, this could lead readers to question whether some datapoints were inadvertently excluded from the analysis.

Moreover, this annotation strategy is not applied consistently. For example, in Supplementary Figure S2, datapoints are not differentiated using *x*, *x'*, and *x''* as in Figures 4–6. Although the caption explains that genotypes are compared within each time point, the visual inconsistency across figures is potentially misleading, and a casual reader might incorrectly assume that all genotypes are compared across all time points.

4. Missing IAA concentration in Supplementary Figure S2

The concentration of IAA used in Supplementary Figure S2 is not stated and should be included.

Reviewer #3

(Remarks to the Author)

The authors have addressed my concerns.

It appears that a paper was published on-line in August 2025 (June preview <https://doi.org/10.1093/plphys/kiaf330>;

<https://pmc.ncbi.nlm.nih.gov/articles/PMC12344411/pdf/nihms-2127192.pdf>), that has alleles for dao2-2 through dao2-5 for single and double dao1 and dao2 lines.

In order to avoid confusion of allele numbers, it appears that the authors should rename their alleles starting at dao2-6.

The phenotypes presented here for the dao1 dao2 lines are consistent with the earlier publication, supporting their data.

Version 2:

Reviewer comments:

Reviewer #1

(Remarks to the Author)

I would like to thank the authors for their detailed and sincere responses to my previous comments. After carefully reviewing the revised manuscript and the rebuttal letter, I am satisfied that all of my concerns have been adequately addressed. The additional data and clarifications have significantly strengthened the manuscript. I have no further comments.

Reviewer #2

(Remarks to the Author)

I would like to first congratulate the authors on this well-executed manuscript and thank the editor for the opportunity to review it. While some questions remain unresolved, this is an expected outcome of a thorough study and reflects the inherent limitations of experimental work. Overall, I believe this revised version has improved compared to the initial submission. I have included only minor comments, which I leave to the editor and authors to evaluate in terms of their relevance.

I have inserted comments directly alongside the authors' rebuttals, indicated by a dash (–). Points not explicitly addressed below are considered resolved from my perspective.

1. We have found that the section describing the generation of this double mutant was incomplete, reason why it looked like we have obtained new alleles of these genes. Let us clarify this:

Original text said: "Previously, ...". Corrected text now says: "Previously, ... by crossing the ...".

Even though it was originally explained in the methods section (line 591), we realized this initial explanation was somehow misleading. Therefore, there is no new alleles involved because both mutants were already described.

– I appreciate the authors' clarification in the main text.

2. We have checked anyway the expression of both genes in the ugt74d1 ugt84b1 background. Since the ugt74d1 is an insertional allele, we expected a null detection of the transcripts (Rebuttal Figure 1a). However, since UGT84B1 is a CRISPR/Cas9-deletion, the expression levels were comparable (very low, anyways) between wild-type and mutant (Rebuttal Figure 1b). In our view, this confirmatory data does not provide anything new to the data, reason why we prefer not to include it.

– I agree that these data may not add substantial new insight; however, similar concerns could arise for other readers, and inclusion of this figure could be helpful. I leave this decision to the editor.

3. Since the only statistically significant phenotype observed in this genotype is on the hypocotyl length, we have included a picture of the hypocotyls of a wild-type plant and a ugt74d1 ugt84b1 in the Figure S1.

– If the image is representative, the difference in hypocotyl length relative to wild type is not readily apparent, in contrast to the quantitative data shown in Figure 3c. While the close-up view is useful, it is unclear why only hypocotyls are shown rather than whole seedlings, as in other genotypes in the same figure. This inconsistency could raise concerns and would benefit from clarification. Again, I leave this decision to the editor.

4. Even though we found a typically IAA-related phenotype (hypocotyl length) affected in the double mutant, we agree the effects are subtle, so we rephrase the sentence:

"A major contribution of the present work concerns the functional importance of the IAA glycosylation pathway, conventionally overlooked mainly due to the absence of obvious phenotypes in single and doubles ugt mutants^{14-16,40}".

– I agree with the revised wording. It may also be appropriate to reference Figure 3 here, as the data support this statement.

5. We acknowledge it may look a bit arbitrary, but we decided to use the same IAA concentration as in Hayashi et al. 2021

for the cotreatment with KKI. We now specify that Figure S2 experiment was carried out using 1 μ M of labelled IAA.

– The manuscript would benefit from including this clarification in the main text to improve transparency regarding experimental conditions.

Reviewer #1

In this study, the authors generated a comprehensive set of combinatorial mutants of IAA-inactivating enzymes, including GH3s, DAOs, and UGTs, and then analyzed their resulting phenotypes. The de novo synthesis and steady-state levels of IAA metabolites in these mutants were measured using state-of-the-art LC-MS/MS analysis with isotope-labeled internal standards. This study extensively investigates the metabolic profiles of IAA metabolites at both de novo synthesis and steady-state levels. Finally author performed RNA-seq analyses among *gh3oct*, *gh3oct dao1* and WT+IAA plants.

However, I have several concerns on the genetic materials and statistical analyses used in this study, which are detailed below. These concerns cast serious doubt on the conclusions of this manuscript.

Thank you for your thoughtful evaluation of our work. We have addressed all your points. We hope the new clarifications can solve your concerns.

Major Concerns

1. The *dao2-1* allele (Line 140) The authors used the *dao2-1* allele to generate the *dao1 dao2* double mutant. Zhang et al. (PNAS, 2016; PMID: 27651492) previously reported that this allele contains a T-DNA insertion in the second intron of the *DAO2* gene and that *dao2-1* plants still express *DAO2* at approximately 39% of wild-type levels. This raises a critical question: Is *dao2-1* a true loss-of-function mutant? The residual gene expression suggests it may not be a knockdown allele. Have the authors confirmed the *DAO2* expression level in their *dao1 dao2* double mutants to ensure it is a functional knockout? This is crucial for the interpretation of the phenotypic and metabolite analyses of *dao2-1*, and *dao1 dao2* mutants

We agree with the reviewer that *dao2-1* is likely a knock-down rather than a knock-out. However, we consider this does not change any of our interpretations since our approach has produced two new mutant versions of *DAO2* and they behave in the same fashion.

To facilitate this analysis and to deal with comments regarding the clarity of Figure 2, we have now created a new Figure 2 which only includes details of the mutants analysed and how they have been generated. In this (new) Figure 2, we can see that while the double *dao1-4* (first three *DAO1* alleles were already reported) *dao2-1* does contain the likely leaky allele of *DAO2*, other combinations such as the large deletion encompassing *DAO1 DAO2*, now named *dao1-5 dao2-2*, induced in *ugt74d1* background or the mutation in *DAO2* generated in *gh3oct* background (*gh3oct dao2-3*) have a similar behaviour:

- Phenotypically, *dao1-4 dao2-1* is only statistically different in terms of primary root length (according to the new ANOVA analyses). In the same analyses, *ugt74d1 ugt84b1* do not show differences. When combined in a quadruple mutant removing the function of both *DAO1* and *DAO2* (*dao1-5 dao2-2 ugt74d1 ugt84b1*), these are phenotypically identical to the *dao1-4 dao2-1*. We therefore conclude that the contribution of these 39% of potential wild-transcripts from *DAO2* in the *dao1-4 dao2-1* is unable to mitigate any biological consequence and so can be considered as equivalents.
- Isotope labelled feeding experiments. When we consider the identical genotypes above-mentioned (Figure 4), we can see an equivalent metabolic response in both sets of *dao1 dao2* genotypes, with a major canalization of the exogenously applied auxin towards IAA-Glu and IAA-Asp and the inability to form oxIAA, and oxIAA-Glu, and oxIAA-Asp.

Overall, even if *dao2-1* is a partial loss-of-function, the contribution of the remaining activity is irrelevant to the phenotypes and the metabolic fate of the auxin treatments.

2. Statistical Analysis of Phenotypic Data (Lines 170, 178; Figure 2) The authors analysed the phenotypic data in Figures 2B-2D using a Student's t-test to determine if differences between two groups are statistically significant. This is an inappropriate statistical method for an experimental design that involves comparisons across multiple groups. The authors should re-analyse this data using an Analysis of Variance (ANOVA) followed by a suitable post-hoc test for multiple comparisons, such as Tukey's HSD or Dunnett's Test. Consequently, the conclusions drawn in lines 170 and 178 regarding the hypocotyl lengths of *dao2-1* and the *ugt84b1 ugt74d1* double mutant must be carefully reconsidered in light of a correct statistical analysis.

We are genuinely thankful to the reviewer for this comment, which also aligns with a comment from reviewer 2 so please consider the following as a response to both comments. We agree that the initial statistical comparisons were not carried out properly. Following Reviewers advice, we have now conducted an ANOVA test followed by a Tukey's test for the analyses of these multiple datasets. Results are shown in the (new) Figure 3 and described and discussed in the lines 163-187. As a summary, single *daos*, multiple

*ugt*s and *gh3oct* have no effect on primary root while *dao1 dao2*, *dao1 dao2 ugt84b1* and *dao1 dao2 ugt74d1 ugt84b1* show enhanced primary root growth. In contrast, adding *dao1* or *dao2* mutations to the *gh3oct*, significantly reduces their primary root growth. For the root branching, the landscape is similar although in this case all the strong phenotypes consist in more branching. For the hypocotyl length, while the differences to Col-0 in multiple *daos* and *ugt*s and their combinations are rather mild, there are some combinations with a significant difference. Nevertheless, the stronger phenotypes are still the *gh3oct* with or without *dao* mutations.

Furthermore, the quantitative data of metabolites presented in Figures 3-5, S1, and S2 appear to lack any statistical analysis. All quantitative data throughout the manuscript must be analyzed with appropriate statistical methods, and the results of these analyses should be clearly reported in the table, figures, supplementary table or text.

We are also really thankful to the reviewer for this comment. We have applied the same statistical analysis (ANOVA test followed by a Tukey's test) to the feeding experiments (Figure 4-6, S2) and steady-state levels quantification (Figure S3) as well as to the KKI treatment analysis (new) Figure S4.

3. The *gh3oct* mutant (Lines 285-291) The *gh3 oct* mutant used in this study was previously generated by the same first author (Casanova-Sáez et al., *New Phytologist*, 2022). This paper have demonstrated that IAA-Glu levels were not reduced in the *gh3 oct* mutant. Consistent with the previous paper, the current manuscript shows that IAA-Glu accumulates in the *gh3oct dao1* mutant (Figure 4C, S2B), while IAA-Asp does not. The T-DNA insertion in the *gh3-17* allele (SALK_050597), a component of the *gh3oct* mutant, is located at the C-terminus of the last exon. This position raises the possibility that a truncated, partially functional protein is produced. This would mean the *gh3oct* mutant is not a null (complete loss-of-function) mutant for Group II GH3s. This concern is supported by another study (Guo et al., *BBRC*, 2022), which generated a *gh3oct* mutant using CRISPR/Cas9. Their mutant exhibited much more severe high-auxin phenotypes (e.g., strong root growth inhibition, more lateral and adventitious roots) than the one used in this work. The phenotypes reported by Guo et al. are also consistent with those induced by chemical GH3 inhibitors (Xie Y, *PNAS* 2022: PMID: 36454752, Fukui K, *PNAS* 2022: PMID: 35914172). This leads to a key question: Is the *gh3oct* mutant used in this study resistant to the GH3 inhibitor kakeimide (GH3 inhibitor used in this study)? If the *gh3oct* mutant is indeed not a null mutant, all claims regarding the function of Group II GH3s based on this genetic background should be carefully revised and toned down. The authors acknowledge this possibility in lines 285-291, but the potential impact of this issue on the paper's conclusions needs to be more thoroughly addressed throughout the manuscript.

Again, this was a wonderful suggestion. We have taken a closer look at the potential leaking of this mutant. Contrary to *dao2-1* which seems irrelevant to our conclusions, we have found that the *gh3oct* may have some leaking transcripts (not much though) that potentially retain some functionality of *GH3.5* and *GH3.17*. Since we have RNA-seq data of (+/- IAA) for Col-0, *gh3oct* and *gh3oct dao1-6* we could have a look at the transcriptional landscape of these genes in the wild-type and mutant contexts.

The *gh3oct* combines 8 insertions in the eight group II *GH3* genes. Insertions (either T-DNAs or other forms of similar transference DNA) can generate different changes in the gene functionality. The first one, is just interrupting the natural transcription of the gene. We see this in *GH3.1* (new Figure S6a), where there is transcription at both sides of the insertion although there are no wild-type transcripts. By standard RNA-seq analysis, it seems that *GH3.1* is even upregulated in the *gh3oct dao1* because the gene has not lost their IAA-responsiveness. Another usual scenario for chromatin regions with T-DNAs is the silencing of the region. Normally the small RNA machinery identifies these regions as potentially harmful sequences that have to be converted into heterochromatin. This is what is seen at different extents in *GH3.2*, *GH3.3*, *GH3.6*, and *GH3.9* (Figure S6b,c,f, g). Either before or after the insertion there seems to be a full lack of transcription, likely driven by the heterochromatinization of the region. In the tissue we used for this experiment, *GH3.4* is not expressed under any condition of genotype (Figure S6d).

However, two of the insertions may become problematic since they are at the very end of the genes: *GH3.5* and *GH3.17*. While both insertions have generated a silencing effect that can be easily observed in the signal intensity with or without auxin, it is also true that, potentially, there is a long-enough transcript to generate some wild-type activity. Considering this, we performed a KKI treatment on *gh3oct* plants consistently finding that they are not insensitive. Therefore we concluded that some GH3 activity remain in this background. We have discussed all these new analyses (lines 357-381, 534-538) toned down our conclusions accordingly, and prepared a new Figure (S6) to show all these new results.

4. Metabolite profiling in the *ugt74d1 dao1 dao2* mutant (Lines 231-238) The metabolic profile of oxIAA and oxIAA-amino acid conjugates in the *ugt74d1 dao1 dao2* mutant seems to be very surprising. The authors suggest this points to "possible involvement of additional IAA oxidase activities" and a "complexity and flexibility of the IAA inactivation network higher than currently understood." However, the data in Figures 3, S1, and S2 and previous papers suggest that DAO1 is responsible for almost all oxidation of IAA-amino acid conjugates. The presence of these metabolites in a triple mutant background is therefore unexpected. Have the authors confirmed that the *dao1* and *dao2* mutations are homozygous in their *ugt74d1 dao1 dao2* line? Genotyping confirmation is essential to rule out the possibility of genetic segregation or contamination.

We agree that the profile shown by the triple mutant was interesting and which is why highlighted in the first version. However, steady-state levels of IAA-Glu, IAA-Asp, oxIAA or oxIAA-glc are consistent with these plants being homozygous for *dao1 dao2* (Figure S3). Nevertheless, we re-checked the zygosity of these mutations and included a new set of primers required to do so in the Supplemental Table S1. It requires further experimental approaches to figure out what exactly is behind this behaviour but we consider that it is beyond the scope of this work.

Specific points
in Fig1A upper box.
dao1 dao2 > *dao1* ?

We assume reviewer refers to the Figure 2A. In the very upper box, the information says that there is a nucleotide deletion, which causes a protein frame shift which (putatively) produce a number of amino acids, written in red, before a premature stop codon. In any case, this Figure has changed in the current version.

Figure S2. IAA-Asp

level in *ugt74b1 dao1 dao2* is missing,

We thank the reviewer for this comment. Values were plotted but by accident not shown. We corrected the scale of the Y-axis in order to show the values.

Reviewer #2

As sessile organisms, plants rely on an extensive set of regulatory molecules among which phytohormones are a crucial to adapt their life cycle in response to changes in the environment. Auxin, whose major active form is indole-3-acetic acid (IAA), is a phytohormone involved in nearly every aspect of plants' life, from embryo development to senescence. Therefore, proper spatiotemporal patterns of auxin biosynthesis, transport, catabolism and response are essential for plants' phenotypic plasticity and fertility. Recently, IAA catabolism through the oxidative pathway has been revised placing GH3 and DAOs activity in the same pathway. However, IAA inactivation through glycosylation, catalyzed by UDP-glycosyltransferases, is another major IAA inactivation pathway. In this study, the authors endeavor to examine the interaction between these pathways compared to their individual roles, which is a major contribution in the auxin field.

In general, the paper is logically structured and reads well. I appreciated the conciseness of the introduction and the technical thoroughness of the metabolic side of the study. While reading the manuscript I was both excited—for the beauty of results presented in it and how the authors described them—and overwhelmed by the magnitude of the study. The authors did a great job driving the reader through the manuscript to highlight the main findings, but it is still not a quick read. I do not have a definitive solution for it, but I wanted to acknowledge the challenge and make some suggestions that I think may facilitate data interpretation.

General commentaries:

The authors introduce here for the first time a set of newly generated lines with altered auxin inactivation and the measurement of relevant morphometric traits strongly regulated by auxin concentration, such as hypocotyl and root length and lateral root density. However, such measurements were not accompanied by representative pictures of plants. While the analyses performed here relied on the abovementioned traits as a readout of the metabolic differences observed between genotypes, some of these lines are introduced for the first time, so it would be helpful to count on such pictures to visualize other traits associated to changes in auxin concentration described for lines with altered auxin homeostasis, like cotyledon epinasty, root branching, or adventitious root formation, under the growth conditions (and developmental stage) chosen to carry out the metabolic quantifications.

Thank you for your kind words and careful evaluation of our work. We have addressed all your points. We hope the new clarifications can solve your concerns.

We have now included a new supplementary Figure (S1) showing a representative plant of every newly generated line. We consider that the determination of other phenotypic parameters such as adventitious roots, while interesting is beyond the scope of this work.

The statistical analyses were in many cases inadequate for the interpretation of the results that are later discussed. In most cases, pairwise comparisons between a mutant and WT were tested but later discussion covering the significance of differences in some traits between mutants is mentioned without having assessed statistical significance (ANOVA).

This point was raised by the previous reviewer, and we have addressed it in our earlier response.

I found a number of typos, some of which I point out below, but the authors may want to read through the manuscript again to fix them.

Specific commentaries:

Line 53: DAOs are not spelled out as DIOXYGENASE FOR AUXIN OXIDATION for the first time until line 97. However, GH3, UGTs, and IAA are fully spelled out in the abstract.

We have corrected this lack of consistency.

Line 83: Are both UDP-glycosyltransferases and UDP-glycosyl transferases (line 55) correct?

We would like to thank the reviewer for noticing this discrepancy. We use both indiscriminately but after a careful checking, the formally correct way of spelling this term according to the IUPAC-IUBMB is UDP-glycosyltransferases. Therefore, we have corrected it in the abstract.

Line 85: Since it is mentioned that only three UGTs are associated to IAA inactivation in Arabidopsis, and Figure 1 is titled “main pathways for indole-3-acetic acid (IAA) metabolic inactivation in Arabidopsis”, it would be helpful to also place UGT76E5 in the figure.

That is another useful suggestion. It is now included.

Line 97: Following the same rationale, I suggest including DAO2 in Figure 1, specially because *dao2* is one of the stacked mutants.

We have also included DAO2 in the Figure.

Line 113: Figure 1 title is bold but, through the manuscript, other figures only bolded the name.

Although this may change in the eventual final version, we have now bolded the text of all the Figure titles, for consistency.

Line 115: “inactivation define” should be “inactivation defines.”

We have corrected this typo.

Line 165-166: It is not explicitly stated whether “mild auxin-associated phenotypes” refers to phenotypes associated to increased or decreased auxin concentration. Cite a reference showing that increased primary root length and lateral root density have been previously associated as such.

We would like to thank the reviewer for this notice. We have now explicitly said that it refers to mild high auxin phenotypes.

Line 167: DAO 1 should be DAO1 (not separated). Line 168: “...despite DAO1 being the predominant oxidase in root IAA metabolism”: Add reference.

Both have been corrected.

Line 169: If “... DAO2 contributes to regulating hypocotyl elongation” is known from the literature, cite a reference.

The text explains that singles *dao1* or *dao2* do not root or hypocotyl phenotype and refers to the articles where these genes were reported.

Line 170: Was the difference in hypocotyl length observed in *dao2* compared to WT consistently found across experiment repetitions? If so, how would it be explained the “more complex interplay” leading to *dao1 dao2* not showing such difference?

Our new statistical analysis of the morphometric parameters has altered this section extensively, we believe that this largely answers reviewer concerns.

Line 171: *dao1 dao2* should be italicized.

It is now corrected.

Line 178: I think it would be helpful to first mention that *ugt84b1 ugt74d1* does not show any defects in roots to understand why it is remarkable that *dao1 dao2 ugt74d1 ugt84b1* shows more prominent increase in LRD (but not in length) than *dao1 dao2*.

It is now corrected (line 171).

Lines 187-188: “...nonuple mutants displayed a further reduction in PRL and a greater increase in hypocotyl length...” is not statistically supported since only pairwise comparison to WT has been tested (t-test) rather than multiple comparison (ANOVA).

The new statistical analysis included still supports this sentence.

Lines 189-190: To my understanding, this statement should be accompanied by some evidence showing that *gh3oct* is fully knocked out. Otherwise, some GH3 leaky activity might be further disrupted by *dao* mutants. Gene expression analysis from RNAseq would be helpful here.

We have actually performed this analysis of the potential leaking GH3 activity in the *gh3oct* as explained in detail in the previous reviewer response. Since at this moment of the narrative we are simply showing morphological differences, we are happy with the current text since what we suggest is that the stronger phenotypes shown by *gh3 dao1-6* or *gh3oct dao2-3* are due to higher IAA levels (without explaining where it may be derived from) which is still valid even if *gh3oct* is a strong knock-down.

Line 206: Typo: “observed ing *dao1-1*.”

It has been corrected.

Line 210: Typo: “Trunover”

It has been corrected.

Line 218: Given the large amount of data, and in order to help the reader understand the author’s point, it would be very helpful to reword the sentence in such a way that instead of saying “*ugt84b1 ugt74d1 dao1 dao2* mutant had a stronger impact on lateral root density” it specifies whether LRD was higher or lower and compared to what genotype. For example, “*ugt84b1 ugt74d1 dao1 dao2* mutant showed greater LRD than *dao1 dao2* (or *ugt84b1 ugt74d1*)”. Otherwise, the reader needs to go back to the cited figure to remember and it interrupts the flow. Nonetheless, this statement needs to be statistically supported by an ANOVA.

New statistical analysis was added and text was changed accordingly.

Line 221: “Notably, our data showed an identical lower accumulation of [13C6]IAA” is not statistically supported. ANOVA is required for that.

New statistical analysis was added and text was changed accordingly.

Line 318: In the section titled “RNA-seq supports the auxin hyperaccumulation phenotypes in *gh3oct dao1* and implicates non-group II GH3s in the response to high IAA” the authors did not examine the expression GH3s and DAOs in the *gh3oct*, *gh3oct dao1*, and *gh3oct dao2* backgrounds, which would be essential to understand the degree of disruption that each mutant has in the oxidative pathway.

We have included, and discussed above, a whole new section and Figure S6 explaining the *gh3oct* mutant at transcriptional level. There was already a Figure, now Figure S12, showing the expression levels of the rest of the GH3s. We question the utility of measuring the expression levels of *DAO1* or *DAO2* in *gh3oct dao1-6* or *gh3oct dao2-3* given that the CRISPR/Cas9-induced editions affect the protein frameshift rather than the expression level as T-DNAs do.

Line 362: The 64% is explained only by the combination of PC1 and PC2 (the PCs with the greatest percentage), but the summatory of all PCs surely adds up to more than that.

The aim of the PCA analysis displayed in (new) Figure 7c is to visually show that the two main components of their differential expression behaviour correspond to first (PC1) the genotype (two clear patterns Col-0 vs mutants), and second (PC2) the treatment (top clustered dots corresponding to mock, bottom dots corresponding to IAA samples). While it is true that around one third of the variability is unexplained by either of these components, the point was to emphasize the great contribution of these two main factors.

Line 373: I find counterintuitive that the auxin biosynthesis-related TAA1 (cluster 4) is induced in response to IAA in the IAA-inactivation-defective *gh3oct dao1* mutant. How would the authors explain that?

That is indeed an excellent observation. In 2016, when our lab reported the first study on DAO enzymes in PNAS, we also published a mathematical framework which proved that the phenotypes shown by the

mutants could only be explained by a simultaneous stimulation of IAA biosynthesis (Mellor et al. 2016; <https://doi.org/10.1073/pnas.1604458113>). This work showed the interconnection between the routes and highlighted the role of modelling in order to decipher such complex biological feedback. We have now mention this and cited this work in line 432.

Line 447-448: The overlap (238 genes) is 40% of WT + IAA genes but only ~10% of *gh3oct*-harboring mutants. Considering the number of differentially expressed genes in *gh3oct*- harboring mutants (2000+ genes) and only ~10% overlapping with WT + IAA, I would think that the alteration in their transcriptional programs due to IAA accumulation is not the most prominent factor. Therefore, I find that “largely mimic IAA-treated wildtype plants” is an overstatement.

We appreciate this perspective. However, our interpretation is that while the DEGs found in Col-0 + IAA derived from a short IAA treatment (4 hours in 1 μ M IAA), the DEGs found for the *gh3oct* and *gh3oct dao1-6* are the result of the plant dealing with a long-term IAA accumulation, therefore triggering downstream adaptive responses to manage this alteration. To address this concern, we have changed the term “largely” to “considerably” (line 520).

Line 460: “involving alternative or yet unidentified pathways”: in the introduction, the authors mention IAA methylation (meIAA) as an inactivation pathway. Would it be possible to reanalyze their results to determine meIAA concentration? Also, could the authors show the expression of UGTs in *gh3oct* or *gh3oct dao* mutants?

We would like to thank the reviewer for this suggestion. Indeed, it would require a new set of experiments to determine the meIAA levels which we consider beyond the scope of this work. From our experience, this metabolite is typically detected at very low levels.

We have retrieved the expression levels of *UGT74D1*, *UGT84B1* and *UGT76E5*:

The enhancement of IAA-glc in *gh3oct* or *gh3oct dao1-6* does not seem to be explainable by the transcriptional triggering of the genes. Therefore, to avoid confusing the reader, we consider it better to omit this from the manuscript.

Figure commentaries:

Figure 2

I find panel A confusing and difficult to interpret: the boxes that amplify the DNA/aminoacid fragments that have been modified zoom out from a region that is not clear what region exactly is being amplified. In the spirit of simplifying the message, the authors may want to consider separating the sequence information (maybe as a supplementary figure?) from the structural information (in main figure). For example, placing the WT structure of DAO1 (dark grey)/DAO2 (light grey) on top, below it, the *dao1*, *dao2*, or *dao1 dao2* structures shading in semitransparent black the disrupted parts of the genes in each background. This is just a format suggestion, but the authors may consolidate it otherwise; I just find panel A busy and unclear. Since the scale bar is explicit in panel A, I do not see the need to mention it in the caption. Panels B-D are clear and easy to read. In the caption, they are cited as phenotypic parameters, but these are variables, not parameters. To avoid confusion, they can be referred to as morphometric traits. At the same time, as a colorblind person, it is hard for me to tell apart the colors. I do not recommend combining different shades of green and orange in the same plot. I usually try to combine the three primary colors (red/magenta, yellow, blue/cyan) with black, white, and grey, in addition to symbols if needed to scale up the number of unique

signatures. In panels B-D, I personally don't see the need to color-code the genotypes since they do not add any information and some of them are hardly distinguishable for me: *dao1-1* vs *ugt74d1 ugt84b1*, *dao1 dao2* vs *gh3oct dao1*, or *ugt74d1 dao1 dao2* vs *gh3oct*. I do see the consistency in color usage for specific genotypes across the manuscript. Therefore, if the authors want to stick to color-coding the genotypes (it might be informative for other non-colorblind fellows), I recommend using other colorblind-friendly selection of colors, or adding different shapes (circle, triangle, asterisk) to more "similar" colors. To my understanding, and according to the statement in lines 144 and 151, followed up by that in line 193, the quantitative nature of Figure 2 is to examine the interaction between the conjugation and oxidation pathways for IAA inactivation. Therefore, pairwise comparison to WT (Student's t-test) is not adequate for this purpose and, instead, one way-ANOVA should be performed. For example: are the differences in hypocotyl length and lateral root density observed in *dao1 dao2* significantly different from those observed between *ugt74d1 dao1 dao2* or *ugt74d1 dao1 dao2*? How about the differences in primary root length observed between *ugt84b1 dao1 dao2* and *ugt74d1 ugt84b1 dao1 dao2*? Finally, the methods section describes that "plants were kept in vitro for a maximum of 2 weeks". While I think it is ok for the methods section, it is necessary to specify in Figure 2's caption the seedlings' age in the reported experiment for the sake of reproducibility.

We really appreciate the detailed analysis of this Figure. We have now introduced the following changes:

- We split the (former) Figure 2 into two Figures. The (new) Figure 2 which now details all the mutations and their effects and the (new) Figure 3, which have the phenotypic information.
- As suggested by Reviewer 1 and 2, we have analysed the phenotypic information with an ANOVA and Tukey's test.
- Figure caption now says "Morphometric traits of the IAA metabolism mutants".
- We now say "phenotypic variables" rather than not "parameters".
- We indicate the age of the plants when the phenotype was analysed.

Figure 3 & Supp. Figure S1

Statistical tests are missing, but it would be possible to visually assess the statistical significance by substituting Standard Error by Confidence Intervals 95% and looking at overlapping bars. How do the authors explain that *dao1 dao2* mutant has intermediate evolution of $^{13}\text{C}_6\text{IAA-glc}$? Considering the disruption of IAA-aa oxidation, the expectation would be a compensation through the glycosylation pathway, similarly to what happens in *ugt84b1 ugt74d1* mutant and the GH3/oxidative pathway.

We have now introduced statistical analysis in all the Figures. Regarding the $^{13}\text{C}_6\text{IAA-glc}$ in *dao1 dao2*, our results are consistent with previous reports (Porco et al 2016). We think GH3 conjugation is the fastest and preferred pathway taken by the cell while glycosylation may be slower.

Figure 4

Line color is hard to distinguish between *gh3oct* and *gh3oct dao2* (consider using black). If WT values are the same as those used in Figure 3, it should be mentioned.

We now say that the wild-type levels are the same as in the previous Figure. We are satisfied with the colour code chosen.

Figure 6

Panel A: Number of genes in Venn's diagram are too small to read well. I am aware that now that papers are digital we can zoom in, but the smallest font size should be 6 pt, being 8 pt the recommended font size.

We agree with the reviewer, and we have corrected this. Now numbers are 8 pts size.

Supp. Figure S2

Statistical tests are missing. It is a major drawback that [IAA-glc] is missing. Also, growth conditions to measure metabolite levels in steady-state (5-day old) are different from those used in Figure 3 & Supp. Figure S1 (7-day old). Considering that this study examines IAA deactivation through the interaction of glycosylation and oxidation pathways, and that Figure 3 shows that *dao1 dao2* mutant is impaired in IAA glycosylation, would it be possible to quantify IAA metabolites again to successfully detect IAA-glc (ideally, in 7-day old seedlings to compare to the observations from Figure 3 & Supp. Figure S1)?

Unfortunately, to determine the IAA-glc in these genotypes would require a new round of experiments. Since none of the authors is currently working in this project any longer, it would be very hard to arrange.

Supp. Figure S5

Letters are too small and what bubble represents what genotype is not clear.

We agree with the reviewer, and we have corrected this. Now numbers are 8 pts size and genotypes are assorted more clearly.

Reviewer #3

The authors present an interesting study of various fates of radiolabeled IAA with higher order mutants in the different metabolic pathways. The authors found some interesting results regarding the role of IAA metabolites in regulation of flowering time via FLC. There are a few questions about the study that I'm sure the authors can address.

We are really grateful for these positive comments on our work. We tried to address all the comments.

1. The dao2-1 allele was not fully described in the text, so I looked it up. It appears in previous publications that it is a knockdown allele and not a knockout allele. If this is the case, what do the authors think about the double dao1 dao2 mutant presented here in a dao2-1 knockdown background?

Thank you for this comment. This point has been addressed in Reviewer 1 comments.

2. Line and 207 “Interestingly, ... oxIAA, oxIAA-glc, and oxIAAaa were still present in both single dao1-1 and double dao1 dao2 mutant plants (Figure S2E-G)...” and Line 228. “Surprisingly, ... de novo formation of both oxIAA and oxIAA-aa”. It appears that the dao2-1 knockdown allele has active oxidation activity in dao2-1 mutant. While it is possible that a de novo oxidation activity is present as the authors suggest, what do the authors think about a simpler explanation that the observed oxidative metabolites were from the activity of the knockdown dao1-2 allele?

Thank you for this comment. This point has been addressed in Reviewer 1 comments.

3. Of the genes upregulated in the RNAseq analysis, which gene have auxin regulatory elements in their promoters? This information would point to a hierarchy of direct vs indirect effects of ectopic auxin or auxin metabolite accumulations of the higher order loss of function mutants and exogenous IAA treatments.

We have analysed the cis-elements present in the different clusters shown in Figure 7d. Indeed, cluster 4 (those upregulated only in the octuple and nonuple mutants upon auxin treatment) showed the highest percentage of genes with auxin response elements, further supporting our idea of the excess of auxin as the cause for the triggering of these genes. This new data is discussed in lines 433-439 and is available in Data S1c.

The text should be reviewed for spelling and grammar.

Thank you for this comment. We have tried to fix all the text issues.

Reviewers' comments:

Reviewer #1 (Remarks to the Author):

The authors have done an excellent job addressing the reviewers' concerns, and the manuscript has been substantially strengthened by the addition of new data and clarifications. However, two specific points remain that, in my view, require further attention to ensure the accuracy of the work.

(1) Metabolite profiling in the *ugt74d1 dao1 dao2* mutant (Lines 231-238)

Second-round review comments by reviewer #1

Revised version [line 239-241]: This further suggests the possible involvement of additional AA oxidase activities, and points to a complexity and flexibility of the IAA inactivation network higher than currently understood.”

>While the authors re-checked the zygosity and updated Supplemental Table S1 with the primers used, the results remain puzzling. I am not fully convinced by the differences in oxIAA, oxIAA-aa, and oxIAA-glc profiles between steady-state conditions and feeding experiments in the *ugt74d1 dao1 dao2* mutants. Specifically, the statement in Lines 239-241 should be removed or moved to the discussion section, as the authors themselves admit that explaining this behavior is beyond the scope of the current study. Furthermore, the similar *dao* mutant, *ugt84b1 dao1-5 dao2-2* in Figure S2 exhibits a profile identical to the *dao1-1* mutant, which contradicts the authors' current interpretation in the Results section.

We understand the concern of the reviewer, so we have removed the sentence to avoid reader's confusion.

Second-round review comments by reviewer #1

2. GH3 Activity and Citations (Lines 450-454)

Revised version [Line 450-454]: Notably, recombinant group III GH3 isoforms, mainly GH3.12, and GH3.15 and to a minor extent GH3.14, exhibited IAA-conjugating activity with Glu (21), suggesting that the residual formation of IAA-Glu observed in the *gh3oct* mutant (19) (Figure 5c, Figure S3b) could be ascribed to the activity of these non-group II GH3 members rather than the limited and truncated transcripts of GH5 or GH3.17.

>In Lines 453-454, the authors discuss the involvement of non-group II GH3 members in the residual formation of IAA-Glu in the *gh3oct* mutant. However, there are issues with the evidence provided:

1) Reference 21 is inappropriate here, as the IAA-conjugating activities of GH3.12, 14, and 15 were not investigated in that study.

We have slightly re-phrase the sentence (line 449) but our original statement that “*recombinant group III GH3 isoforms, mainly GH3.12, and GH3.15 and to a minor extent GH3.14, exhibited IAA-conjugating activity with Glu*” is based on Supplementary Dataset 1 of Reference 21, where the IAA-conjugating activity of recombinant non-group II GH3 proteins was experimentally assessed.

Here is the data from that work:

		Group III					
GH3	Sample	pmol/ml					
		IAA-Asp			IAA-Glu		
AtGH3.12	mock_1	<LOD	<LOD		<LOD	<LOD	
	mock_2	<LOD	<LOD		<LOD	<LOD	
	mock_3	<LOD	<LOD		<LOD	<LOD	
	1 mM IAA_1	<LOD	<LOD		2585.8	4409.41 ±	1296.34
	1 mM IAA_2	<LOD	<LOD		5483.9		
	1 mM IAA_3	<LOD	<LOD		5158.6		
AtGH3.14	mock_1	<LOD	<LOD		<LOD	<LOD	
	mock_2	<LOD	<LOD		<LOD	<LOD	
	mock_3	<LOD	<LOD		<LOD	<LOD	
	1 mM IAA_1	<LOD	<LOD		12.5	9.83 ±	3.84
	1 mM IAA_2	<LOD	<LOD		12.6		
	1 mM IAA_3	<LOD	<LOD		4.4		
AtGH3.15	mock_1	<LOD	<LOD		<LOD	<LOD	
	mock_2	<LOD	<LOD		<LOD	<LOD	
	mock_3	<LOD	<LOD		<LOD	<LOD	
	1 mM IAA_1	<LOD	<LOD		37.3	36.34 ±	0.68
	1 mM IAA_2	<LOD	<LOD		35.7		
	1 mM IAA_3	<LOD	<LOD		36.1		

2) Comprehensive enzyme kinetics for GH3.15 (e.g., J. Biol. Chem. 2018, 293:4277-4288) clearly demonstrate that IAA is not a substrate for GH3.15.

The context in lines 450-454 should be removed or moved to the Discussion section. If moved, the authors must provide correct citations that specifically support the IAA-conjugating activity of non-group II GH3s. The current claim that GH3.12, 14, and 15 contribute to IAA-Glu formation is not supported by the cited literature or known biochemical data.

Ref (21) Brunoni, F. et al. Amino acid conjugation of oxIAA is a secondary metabolic regulation involved in auxin homeostasis. *New Phytol* 238, 2264-2270, doi:10.1111/nph.18887 (2023).

The findings reported for *Arabidopsis* GH3.15 and IAA in Reference 21 are consistent with those shown in Figure 1A and Table 2 of Sherp *et al.* (J. Biol. Chem. 2018, 293:4277–4288), where the authors demonstrated that “*AtGH3.15 used IAA, but less efficiently than IBA*”. Notably, although different experimental approaches were employed, both studies yielded comparable conclusions. In Sherp *et al.*, enzymatic activity was measured spectrophotometrically using a coupled assay system that monitors the first half-reaction (substrate adenylation). In contrast, Brunoni *et al.* quantified enzymatic activity by direct LC–MS detection of the IAA–amino acid conjugate, thereby assessing completion of the full catalytic reaction. Importantly, a more recent study by Holland and Jez (J. Biol. Chem. 2024, 300.7) demonstrated that, for certain substrates and GH3 isoforms exhibiting relaxed substrate specificity, the first half-reaction (formation of the substrate–AMP intermediate) can proceed independently of the second half-reaction. In such cases, adenylation occurs without subsequent formation of the final amino acid conjugate. This distinction underscores the methodological differences between assays that monitor only adenylation versus those that directly detect the conjugated product.

Reviewer #2 (Remarks to the Author):

I appreciate the authors' thorough revision of the manuscript and their clear effort to address the concerns raised in my previous review. The manuscript has improved considerably, particularly with respect to data presentation and statistical rigor. However, several important issues remain unresolved and require further attention.

We thank the reviewer for his/her positive opinion on our work and hope this version solves all the concerns.

Overall, the remaining concerns are relatively focused and can be addressed by further refining two main aspects of the study: (1) ensuring that the statistical analyses are fully aligned with the experimental designs, and (2) strengthening the characterization of the *ugt74d1 ugt84b1* double mutant.

For the latter, the manuscript would benefit from (a) an expression analysis of UGT74D1 and UGT84B1 in the *ugt74d1 ugt84b1* double mutant—particularly given that a new mutant allele was generated in this work—and (b) inclusion of representative images of this newly generated mutant to facilitate phenotypic interpretation.

We have found that the section describing the generation of this double mutant was incomplete, reason why it looked like we have obtained new alleles of these genes. Let us clarify this:

Original text said:

*“Previously, we identified UGT84B1 and UGT74D1 as the main UDP-glycosyltransferases responsible for conjugating glucose to both IAA and oxIAA14. To effectively disrupt the IAA glycosylation pathway, we generated a *ugt84b1 ugt74d1* double mutant by CRISPR/Cas9-mediated knock-out of UGT84B114 in the T-DNA *ugt74d1* mutant background¹⁵”.*

Corrected text now says:

*“Previously, we identified UGT84B1 and UGT74D1 as the main UDP-glycosyltransferases responsible for conjugating glucose to both IAA and oxIAA14. To effectively disrupt the IAA glycosylation pathway, we generated a *ugt84b1 ugt74d1* double mutant **by crossing the** CRISPR/Cas9-mediated knock-out of UGT84B1¹⁴ in the T-DNA *ugt74d1* mutant background¹⁵”.*

Even though it was originally explained in the methods section (line 591), we realized this initial explanation was somehow misleading. Therefore, there is no new alleles involved because both mutants were already described. We have checked anyway the expression of both genes in the *ugt74d1 ugt84b1* background. Since the *ugt74d1* is an insertional allele, we expected a null detection of the transcripts (**Rebuttal Figure 1a**). However, since *UGT84B1* is a CRISPR/Cas9-deletion, the expression levels were comparable (very low, anyways) between wild-type and mutant (**Rebuttal Figure 1b**). In our view, this confirmatory data does not provide anything new to the data, reason why we prefer not to include it.

Rebuttal Figure 1. Expression levels of UGT74D1 and UGT84B1 in the *ugt74d1 ugt84b1* double mutant. (a) As described in Mateo-Bonmatí *et al.* 2021 (New Phytol), *ugt74d1* T-DNA significantly reduces *UGT74D1* expression levels. (b) However, CRISPR/Cas9 deletion over *UGT84B1* does not affect its expression levels although they remain low and non-functional in the mutant background. In red the deleted region at *UGT84B1*. More details about the edition can be found in Mateo-Bonmatí *et al.* 2021. RNA was

isolated from 3 biological replicates of 7-day old seedlings, analyses in triplicate. RNA was isolated

using Phenol-Chloroform method. In order to detect *UGT84B1*, we used gene-specific primers and SuperScrip IV (ThermoFisher). Expression was normalized to two housekeeping genes *PP2A* and *UBC*. *P* values shown correspond to Student's *t* test comparisons between wild-type and mutant levels.

Major issues

1. Interpretation of phenotypes in *ugt* mutants and missing supporting data

In line 501, the authors state that “a major contribution of the present work concerns the functional importance of the IAA glycosylation pathway, conventionally overlooked mainly due to the absence of noticeable phenotypes in single *ugt* mutants.” However, representative images of the newly generated *ugt84b1 ugt74d1* double mutant are not shown in the newly provided Supplementary Figure S1, whereas images for *dao1-4 dao2-1* and the higher-order *ugt74d1 ugt84b1 dao1-5 dao2-2* mutant are included.

Since the only statistically significantly phenotype observed in this genotype is on the hypocotyl length, we have included a picture of the hypocotyls of a wild-type plant and a *ugt74d1 ugt84b1* in the Figure S1.

In Figure 3, the double mutant shows a small but statistically significant increase in hypocotyl length relative to WT, while no differences in root length or lateral root formation are reported. In the absence of additional phenotypic documentation, it would be more accurate to state that both single and double *ugt* mutants lack obvious auxin-related phenotypes, unless the authors provide visual or quantitative evidence for additional traits that were not measured.

Even though we found a typically IAA-related phenotype (hypocotyl length) affected in the double mutant, we agree the effects are subtle, so we rephrase the sentence:

“A major contribution of the present work concerns the functional importance of the IAA glycosylation pathway, conventionally overlooked mainly due to the absence of obvious phenotypes in single and doubles ugt mutants^{14-16,40}”

In light of the results for *gh3oct*—which, despite undetectable conversion of IAA to IAA-Glu and IAA-Asp, shows normal sensitivity to KKI and only weak auxin-related phenotypes (potentially explained by the newly disclosed leaky expression of some GH3 genes)—it is reasonable to ask whether a similar explanation could apply to the *ugt74d1 ugt84b1* double mutant. Specifically, residual expression of either gene could potentially account for the lack of strong phenotypes despite near-complete loss of IAA-Glc formation.

Given that this double mutant was generated in the present study through creation of a new *UGT84B1* allele, an expression analysis documenting the degree of transcriptional disruption is warranted.

As explained above, this is not the case. I have not generated any new *UGT84B1* allele. Nevertheless, we have included in this letter an analysis of the expression of both genes. As we already reported for *UGT74D1*, the insertion knocks out the gene completely. Also, as one may expect, the expression levels are not affected on a CRISPR/Cas9 mutant although the functionality clearly is **Rebuttal Figure 1**.

2. Inappropriate statistical analysis in Supplementary Figure S4

The statistical analysis used in Supplementary Figure S4 does not match the experimental design. As the experiment follows a Genotype x Treatment structure, a two-way ANOVA should be performed instead of a one-way ANOVA. Additionally, figure captions should be written to allow independent interpretation. In this context, abbreviations such as “KKI” should be spelled out in full.

We thank the reviewer for this comment. We have re-run the analysis with a two-way ANOVA with no variation in the outcome. We have also changed the Figure caption to include the full name of KKI.

3. Statistical analysis in Supplementary Figure S12

Based on the Genotype x Treatment experimental design, the use of a Student's t-test in Supplementary Figure S12 is not appropriate. A two-way ANOVA should be applied instead.

We have applied a two-way ANOVA analysis for this dataset and changed accordingly the Figure, the capture and the related text.

4. Inconsistent IAA concentrations across experiments

Throughout the manuscript, the authors consistently use an exogenous IAA concentration of 1 μM (Figures 4, 5, and 7, and presumably Supplementary Figure S2, although this is not stated). However, a fivefold lower concentration (0.2 μM IAA) is used to assess IAA deactivation dynamics in the *ilr1-1 iar3-2 ill2-1* mutant (Figure 6). I would think that has something to do with a potentially deadly accumulation of IAA in KKI-treated *ilr1-1 iar3-2 ill2-1* mutant or different sensitivity of Ws-2 background, but this should probably be explicitly explained.

We acknowledge it may look a bit arbitrary, but we decided to use the same IAA concentration as in Hayashi et al. 2021 for the cotreatment with KKI. We now specify that Figure S2 experiment was carried out using 1 μM of labelled IAA.

5. Missing statistical analysis in Supplementary Figure S13

No statistical analysis is described for Supplementary Figure S13. Given the experimental design, a two-way ANOVA is expected and should be reported.

We have implemented the same two-way ANOVA as in Figure S12.

Minor issues

1. Consistency in mutant nomenclature

Mutant names should be used consistently throughout the manuscript. For example, the quadruple mutant *ugt74d1 ugt84b1 dao1-5 dao2-2* is referred to as *dao1-5 dao2-2 ugt74d1 ugt84b1* in lines 176–177.

Thank you for noticing this inconsistency. It has now been corrected.

2. Role of MeIAA in compensatory IAA inactivation

In the rebuttal letter, the authors state that experiments measuring MeIAA levels are beyond the scope of the study. I respectfully disagree with this assessment. The Introduction explicitly discusses IAA inactivation via methylation, and in lines 531–532 the authors propose that metabolic compensation may occur through redirection of IAA into alternative inactivation pathways, potentially including unidentified routes.

Despite this framing, the manuscript does not assess whether MeIAA contributes to such compensation. In this context, the rebuttal response reads as somewhat generic, particularly given the manuscript's stated goal of providing a "comprehensive characterization of IAA inactivation pathways."

That said, while inclusion of MeIAA measurements would strengthen the study, I recognize the experimental effort required relative to the incremental insight gained, and therefore I do not insist further on this point.

We appreciate the reviewer's understanding in this regard.

3. Inconsistent annotation of statistical significance

The annotation of statistical significance groups is inconsistent across figures and may be confusing for readers. Standard practice is to annotate each datapoint with its corresponding significance group. In Figures 4–6, however, each group is labeled with a single letter, requiring the reader to infer datapoint membership based on proximity. Unless explicitly stated, this could lead readers to question whether some datapoints were inadvertently excluded from the analysis.

Moreover, this annotation strategy is not applied consistently. For example, in Supplementary Figure S2, datapoints are not differentiated using x , x' , and x'' as in Figures 4–6. Although the caption explains that genotypes are compared within each time point, the visual inconsistency across figures is

potentially misleading, and a casual reader might incorrectly assume that all genotypes are compared across all time points.

We now provide as a Supplemental Data set 3 the whole set of raw data so anyone can reproduce our analyses. We have not found a better way to represent this time-course metabolic measurements.

4. Missing IAA concentration in Supplementary Figure S2

The concentration of IAA used in Supplementary Figure S2 is not stated and should be included.

This now has been corrected.

Reviewer #3 (Remarks to the Author):

The authors have addressed my concerns.

It appears that a paper was published on-line in August 2025 (June preview <https://doi.org/10.1093/plphys/kiaf330>; <https://pmc.ncbi.nlm.nih.gov/articles/PMC12344411/pdf/nihms-2127192.pdf>), that has alleles for dao2-2 through dao2-5 for single and double dao1 and dao2 lines.

In order to avoid confusion of allele numbers, it appears that the authors should rename their alleles starting at dao2-6.

The phenotypes presented here for the dao1 dao2 lines are consistent with the earlier publication, supporting their data.

We truly appreciate this comment since this would indeed introduce more noise into the community. We have changed the names of the *dao2* alleles to avoid such confusion.

Reviewers' comments:

Reviewer #1:

I would like to thank the authors for their detailed and sincere responses to my previous comments. After carefully reviewing the revised manuscript and the rebuttal letter, I am satisfied that all of my concerns have been adequately addressed. The additional data and clarifications have significantly strengthened the manuscript. I have no further comments.

We would like to thank Reviewer #1 for her/his valuable comments that definitively helped to improve our manuscript.

Reviewer #2:

I would like to first congratulate the authors on this well-executed manuscript and thank the editor for the opportunity to review it. While some questions remain unresolved, this is an expected outcome of a thorough study and reflects the inherent limitations of experimental work. Overall, I believe this revised version has improved compared to the initial submission. I have included only minor comments, which I leave to the editor and authors to evaluate in terms of their relevance.

We also truly appreciate the time and energy that Reviewer #2 has used to help us to improve our publication.

I have inserted comments directly alongside the authors' rebuttals, indicated by a dash (-). Points not explicitly addressed below are considered resolved from my perspective.

1. We have found that the section describing the generation of this double mutant was incomplete, reason why it looked like we have obtained new alleles of these genes. Let us clarify this: Original text said: "Previously, ...". Corrected text now says: "Previously, ... by crossing the ...". Even though it was originally explained in the methods section (line 591), we realized this initial explanation was somehow misleading. Therefore, there is no new alleles involved because both mutants were already described.

- I appreciate the authors' clarification in the main text.

2. We have checked anyway the expression of both genes in the *ugt74d1ugt84b1* background. Since the *ugt74d1* is an insertional allele, we expected a null detection of the transcripts (Rebuttal Figure 1a). However, since *UGT84B1* is a CRISPR/Cas9-deletion, the expression levels were comparable (very low, anyways) between wild-type and mutant (Rebutal Figure 1b). In our view, this confirmatory data does not provide anything new to the data, reason why we prefer not to include it.

- I agree that these data may not add substantial new insight; however, similar concerns could arise for other readers, and inclusion of this figure could be helpful. I leave this decision to the editor.

Following your advice, we have now included this figure in the Supplemental Figure 1, and mentioned in line 161.

3. Since the only statistically significant phenotype observed in this genotype is on the hypocotyl length, we have included a picture of the hypocotyls of a wild-type plant and a *ugt74d1 ugt84b1* in the Figure S1.

- If the image is representative, the difference in hypocotyl length relative to wild type is not readily apparent, in contrast to the quantitative data shown in Figure 3c. While the close-up view is useful, it is unclear why only hypocotyls are shown rather than whole seedlings, as in other genotypes in the same figure. This inconsistency could raise concerns and would benefit from clarification. Again, I leave this decision to the editor.

Following your suggestion, we have now included a whole seedling picture in the Supplemental Figure 1, and mentioned in line 161.

4. Even though we found a typically IAA-related phenotype (hypocotyl length) affected in the double mutant, we agree the effects are subtle, so we rephrase the sentence:

“A major contribution of the present work concerns the functional importance of the IAA glycosylation pathway, conventionally overlooked mainly due to the absence of obvious phenotypes in single and doubles ugt mutants^{14-16,40}”.

– I agree with the revised wording. It may also be appropriate to reference Figure 3 here, as the data support this statement.

We have revised it accordingly, line 411.

5. We acknowledge it may look a bit arbitrary, but we decided to use the same IAA concentration as in Hayashi et al. 2021 for the cotreatment with KKI. We now specify that Figure S2 experiment was carried out using 1 μ M of labelled IAA.

– The manuscript would benefit from including this clarification in the main text to improve transparency regarding experimental conditions.

This clarification has now been included (line 262).

As sessile organisms, plants rely on an extensive set of regulatory molecules among which phytohormones are a crucial to adapt their life cycle in response to changes in the environment. Auxin, whose major active form is indole-3-acetic acid (IAA), is a phytohormone involved in nearly every aspect of plants' life, from embryo development to senescence. Therefore, proper spatiotemporal patterns of auxin biosynthesis, transport, catabolism and response are essential for plants' phenotypic plasticity and fertility. Recently, IAA catabolism through the oxidative pathway has been revised placing GH3 and DAOs activity in the same pathway. However, IAA inactivation through glycosylation, catalyzed by UDP-glycosyltransferases, is another major IAA inactivation pathway. In this study, the authors endeavor to examine the interaction between these pathways compared to their individual roles, which is a major contribution in the auxin field.

In general, the paper is logically structured and reads well. I appreciated the conciseness of the introduction and the technical thoroughness of the metabolic side of the study. While reading the manuscript I was both excited—for the beauty of results presented in it and how the authors described them—and overwhelmed by the magnitude of the study. The authors did a great job driving the reader through the manuscript to highlight the main findings, but it is still not a quick read. I do not have a definitive solution for it, but I wanted to acknowledge the challenge and make some suggestions that I think may facilitate data interpretation.

General commentaries:

The authors introduce here for the first time a set of newly generated lines with altered auxin inactivation and the measurement of relevant morphometric traits strongly regulated by auxin concentration, such as hypocotyl and root length and lateral root density. However, such measurements were not accompanied by representative pictures of plants. While the analyses performed here relied on the abovementioned traits as a readout of the metabolic differences observed between genotypes, some of these lines are introduced for the first time, so it would be helpful to count on such pictures to visualize other traits associated to changes in auxin concentration described for lines with altered auxin homeostasis, like cotyledon epinasty, root branching, or adventitious root formation, under the growth conditions (and developmental stage) chosen to carry out the metabolic quantifications.

The statistical analyses were in many cases inadequate for the interpretation of the results that are later discussed. In most cases, pairwise comparisons between a mutant and WT were tested but later discussion covering the significance of differences in some traits between mutants is mentioned without having assessed statistical significance (ANOVA).

I found a number of typos, some of which I point out below, but the authors may want to read through the manuscript again to fix them.

Specific commentaries:

Line 53: DAOs are not spelled out as DIOXYGENASE FOR AUXIN OXIDATION for the first time until line 97. However, GH3, UGTs, and IAA are fully spelled out in the abstract.

Line 83: Are both UDP-glycosyltransferases and UDP-glycosyl transferases (line 55) correct?

Line 85: Since it is mentioned that only three UGTs are associated to IAA inactivation in *Arabidopsis*, and Figure 1 is titled “*main pathways for indole-3-acetic acid (IAA) metabolic inactivation in Arabidopsis*”, it would be helpful to also place UGT76E5 in the figure.

Line 97: Following the same rationale, I suggest including DAO2 in Figure 1, specially because *dao2* is one of the stacked mutants.

Line 113: Figure 1 title is bold but, through the manuscript, other figures only bolded the name.

Line 115: “inactivation define” should be “inactivation defines.”

Line 165-166: It is not explicitly stated whether “*mild auxin-associated phenotypes*” refers to phenotypes associated to increased or decreased auxin concentration. Cite a reference showing that increased primary root length and lateral root density have been previously associated as such.

Line 167: DAO 1 should be DAO1 (not separated).

Line 168: “...*despite DAO1 being the predominant oxidase in root IAA metabolism*”: Add reference.

Line 169: If “... *DAO2 contributes to regulating hypocotyl elongation*” is known from the literature, cite a reference.

Line 170: Was the difference in hypocotyl length observed in *dao2* compared to WT consistently found across experiment repetitions? If so, how would it be explained the “*more complex interplay*” leading to *dao1 dao2* not showing such difference?

Line 171: *dao1 dao2* should be italicized.

Line 178: I think it would be helpful to first mention that *ugt84b1 ugt74d1* does not show any defects in roots to understand why it is remarkable that *dao1 dao2 ugt74d1 ugt84b1* shows more prominent increase in LRD (but not in length) than *dao1 dao2*.

Lines 187-188: “...*nonuple mutants displayed a further reduction in PRL and a greater increase in hypocotyl length...*” is not statistically supported since only pairwise comparison to WT has been tested (t-test) rather than multiple comparison (ANOVA).

Lines 189-190: To my understanding, this statement should be accompanied by some evidence showing that *gh3oct* is fully knocked out. Otherwise, some GH3 leaky activity might be further disrupted by *dao* mutants. Gene expression analysis from RNAseq would be helpful here.

Line 206: Typo: “*observed inq dao1-1.*”

Line 210: Typo: “Trunover”

Line 218: Given the large amount of data, and in order to help the reader understand the author’s point, it would be very helpful to reword the sentence in such a way that instead of saying “*ugt84b1 ugt74d1 dao1 dao2 mutant had a stronger impact on lateral root*”

density” it specifies whether LRD was higher or lower and compared to what genotype. For example, “*ugt84b1 ugt74d1 dao1 dao2 mutant showed greater LRD than dao1 dao2 (or ugt84b1 ugt74d1)*”. Otherwise, the reader needs to go back to the cited figure to remember and it interrupts the flow. Nonetheless, this statement needs to be statistically supported by an ANOVA.

Line 221: “*Notably, our data showed an identical lower accumulation of [13C6]IAA*” is not statistically supported. ANOVA is required for that.

Line 318: In the section titled “*RNA-seq supports the auxin hyperaccumulation phenotypes in gh3oct dao1 and implicates non-group II GH3s in the response to high IAA*” the authors did not examine the expression *GH3s* and *DAOs* in the *gh3oct*, *gh3oct dao1*, and *gh3oct dao2* backgrounds, which would be essential to understand the degree of disruption that each mutant has in the oxidative pathway.

Line 362: The 64% is explained only by the combination of PC1 and PC2 (the PCs with the greatest percentage), but the summatory of all PCs surely adds up to more than that.

Line 373: I find counterintuitive that the auxin biosynthesis-related *TAA1* (cluster 4) is induced in response to IAA in the IAA-inactivation-defective *gh3oct dao1* mutant. How would the authors explain that?

Line 447-448: The overlap (238 genes) is 40% of WT + IAA genes but only ~10% of *gh3oct*-harboring mutants. Considering the number of differentially expressed genes in *gh3oct*- harboring mutants (2000+ genes) and only ~10% overlapping with WT + IAA, I would think that the alteration in their transcriptional programs due to IAA accumulation is not the most prominent factor. Therefore, I find that “*largely mimic IAA-treated wild-type plants*” is an overstatement.

Line 460: “*involving alternative or yet unidentified pathways*”: in the introduction, the authors mention IAA methylation (meIAA) as an inactivation pathway. Would it be possible to reanalyze their results to determine meIAA concentration? Also, could the authors show the expression of UGTs in *gh3oct* or *gh3oct dao* mutants?

Figure commentaries:

Figure 2

I find panel A confusing and difficult to interpret: the boxes that amplify the DNA/aminoacid fragments that have been modified zoom out from a region that is not clear what region exactly is being amplified. In the spirit of simplifying the message, the authors may want to consider separating the sequence information (maybe as a supplementary figure?) from the structural information (in main figure). For example, placing the WT structure of DAO1 (dark grey)/DAO2 (light grey) on top, below it, the *dao1*, *dao2*, or *dao1 dao2* structures shading in semitransparent black the disrupted parts of the genes in each background. This is just a format suggestion, but the authors may consolidate it otherwise; I just find panel A busy and unclear. Since the scale bar is explicit in panel A, I do not see the need to mention it in the caption.

Panels B-D are clear and easy to read. In the caption, they are cited as phenotypic parameters, but these are variables, not parameters. To avoid confusion, they can be referred to as morphometric traits. At the same time, as a colorblind person, it is hard for

me to tell apart the colors. I do not recommend combining different shades of green and orange in the same plot. I usually try to combine the three primary colors (red/magenta, yellow, blue/cyan) with black, white, and grey, in addition to symbols if needed to scale up the number of unique signatures. In panels B-D, I personally don't see the need to color-code the genotypes since they do not add any information and some of them are hardly distinguishable for me: *dao1-1 vs ugt74d1 ugt84b1*, *dao1 dao2 vs gh3oct dao1*, or *ugt74d1 dao1 dao2 vs gh3oct*. I do see the consistency in color usage for specific genotypes across the manuscript. Therefore, if the authors want to stick to color-coding the genotypes (it might be informative for other non-colorblind fellows), I recommend using other colorblind-friendly selection of colors, or adding different shapes (circle, triangle, asterisk) to more "similar" colors.

To my understanding, and according to the statement in lines 144 and 151, followed up by that in line 193, the quantitative nature of Figure 2 is to examine the interaction between the conjugation and oxidation pathways for IAA inactivation. Therefore, pairwise comparison to WT (Student's t-test) is not adequate for this purpose and, instead, one way-ANOVA should be performed. For example: are the differences in hypocotyl length and lateral root density observed in *dao1 dao2* significantly different from those observed between *ugt74d1 dao1 dao2* or *ugt74d1 dao1 dao2*? How about the differences in primary root length observed between *ugt84b1 dao1 dao2* and *ugt74d1 ugt84b1 dao1 dao2*?

Finally, the methods section describes that "*plants were kept in vitro for a maximum of 2 weeks*". While I think it is ok for the methods section, it is necessary to specify in Figure 2's caption the seedlings' age in the reported experiment for the sake of reproducibility.

Figure 3 & Supp. Figure S1

Statistical tests are missing, but it would be possible to visually assess the statistical significance by substituting Standard Error by Confidence Intervals 95% and looking at overlapping bars.

How do the authors explain that *dao1 dao2* mutant has intermediate evolution of 13C-IAA-glc? Considering the disruption of IAA-aa oxidation, the expectation would be a compensation through the glycosylation pathway, similarly to what happens in *ugt84b1 ugt74d1* mutant and the GH3/oxidative pathway.

Figure 4

Line color is hard to distinguish between *gh3oct* and *gh3oct dao2* (consider using black). If WT values are the same as those used in Figure 3, it should be mentioned.

Figure 6

Panel A: Number of genes in Venn's diagram are too small to read well. I am aware that now that papers are digital we can zoom in, but the smallest font size should be 6 pt, being 8 pt the recommended font size.

Supp. Figure S2

Statistical tests are missing. It is a major drawback that [IAA-glc] is missing. Also, growth conditions to measure metabolite levels in steady-state (5-day old) are different from

those used in Figure 3 & Supp. Figure S1 (7-day old). Considering that this study examines IAA deactivation through the interaction of glycosylation and oxidation pathways, and that Figure 3 shows that *dao1 dao2* mutant is impaired in IAA glycosylation, would it be possible to quantify IAA metabolites again to successfully detect IAA-glc (ideally, in 7-day old seedlings to compare to the observations from Figure 3 & Supp. Figure S1)?

Supp. Figure S5

Letters are too small and what bubble represents what genotype is not clear.